# Emit now, mitigate later? Earth system reversibility under overshoots of different magnitude and duration

Jörg Schwinger[1], Ali Asaadi[1], Norman Julius Steinert[1], Hanna Lee[1,2]

[1]NORCE Climate & Environment, Bjerknes Centre for Climate Research, Bergen, Norway
[2]Department of Biology, Norwegian University of Science and Technology, Trondheim, Norway

*Correspondence to*: Jörg Schwinger (jorg.schwinger0@gmail.com)

**Abstract.** Anthropogenic $CO_2$ emissions cause irreversible climate change on centennial to millennial time scales, yet current mitigation efforts are insufficient to limit global warming to a level that is considered safe. Carbon Dioxide Removal (CDR) has been suggested as an option to partially reverse climate change and to return the Earth system to a less dangerous state after a period of temperature overshoot. Whether or to what extent such partial reversal of climate change under CDR would happen is, next to socio-economic feasibility and sustainability, key to assessing CDR as a mitigation option. Here, we use a state-of-the-art Earth system model that includes a representation of permafrost carbon to investigate the reversibility of the Earth system after overshoots of different duration and magnitude in idealized simulations. We find that atmospheric $CO_2$ concentrations are slightly lower after an overshoot, compared to a reference simulation without overshoot, due to a near-perfect compensation of carbon losses from land by increased ocean carbon uptake during the overshoot periods. The legacy of an overshoot is, on a centennial time scale, indiscernible (within natural variability) from a reference case without overshoot for many aspects of the Earth system including global average surface temperature, marine and terrestrial productivity, strength of the Atlantic meridional overturning circulation, surface ocean pH, surface $O_2$ concentration, and permafrost extent, except in the most extreme overshoot scenario considered in this study. Consistent with previous studies, we find irreversibility in permafrost carbon and deep ocean properties like sea water temperature, pH, and $O_2$ concentrations. We do not find any indication of tipping points or self-reinforcing feedbacks that would put the Earth system on a significantly different trajectory after an overshoot. Hence, the effectiveness of CDR in partially reversing large scale patterns of climate change might not be the main issue of CDR but rather the impacts and risks that would occur during the period of elevated temperatures during the overshoot.

## 1 Introduction

Although it is still not geophysically impossible to limit global average temperature increase to 1.5°C by immediate and unprecedented emission reductions (IPCC, 2018; Riahi et al., 2021), there is currently no evidence that adequate reductions are being implemented (UNEP, 2021; Friedlingstein et al., 2022). Therefore, it is unlikely that the 1.5°C target laid down in

the Paris Agreement will be met. It might, however, be achieved after a period of temperature overshoot, that is, by compensating too large past and near-term emissions by net negative emissions at a later time. Several methods that could potentially remove $CO_2$ from the atmosphere have been proposed, but there are large uncertainties regarding the effectiveness, feasibility, and sustainability of such mitigation options (Smith et al., 2016; Fuss et al., 2018; Keller et al., 2018b; Lawrence et al., 2018).

From an Earth system perspective, the issue of reversibility (used here to denote a partial reversal of climate change in an overshoot pathway towards an Earth system state in a reference pathway without overshoot) is key in assessing the effectiveness and risks of mitigation pathways that rely on carbon dioxide removal (CDR): Will the climate be in the same or at least in a similar state after a period of temperature overshoot compared to pathways where a temperature target is reached without overshoot? Are there critical limits to the duration and/or magnitude of an overshoot beyond which (aspects of) climate

change become irreversible? Are there tipping points, beyond which self-accelerating feedbacks make a return to a safe climate state impossible or at least difficult?

    Previous studies have suggested that many aspects of the physical climate system are indeed reversible, although in general with some hysteresis behaviour (Boucher et al., 2012; Wu et al., 2015; Tokarska and Zickfeld, 2015; Jeltsch-Thömmes et al., 2020). In model simulations of net negative $CO_2$ emissions, the global mean surface temperature decreases rapidly, while the

slow component of the temperature response (mainly due to thermal inertia of the oceans) is relatively small (Held et al., 2010). Arctic sea ice recovers quickly following the decrease in surface temperature, and the same holds for surface ocean pH, which is tied to the atmospheric $CO_2$ concentration (Boucher et al., 2012; Wu et al., 2015). Precipitation and cloud cover show some hysteresis in their response to negative emissions (Boucher et al., 2012; Wu et al., 2015), but the timescales involved are relatively short. The Atlantic meridional overturning circulation (AMOC) can show an increase above pre-industrial strength

in idealized model simulations when $CO_2$ is returned to pre-industrial levels (Jackson et al., 2014), but this behaviour is model dependent and is not found in simulations with more realistic pathways of CDR deployment (Schwinger et al., 2022).

    Other Earth system processes have been shown to be irreversible on multi-decadal to millennial time scales. These processes include carbon release from permafrost (MacDougall, 2013), thermosteric sea level rise (MacDougall, 2013; Bouttes et al., 2013; Wu et al., 2015; Tokarska and Zickfeld, 2015), ice sheet losses and associated sea level rise (MacDougall, 2013), as

well as changes in ocean temperature, oxygen content and pH (Mathesius et al., 2015; Li et al., 2020).

    Many modelling studies of CDR rely on extremely idealized experiments, where atmospheric $CO_2$ concentration is reduced to pre-industrial levels either abruptly or by some prescribed rate (Cao and Caldeira, 2010; Boucher et al., 2012; Vichi et al., 2013; Bouttes et al., 2013; Wu et al., 2015; Zickfeld et al., 2016; Keller et al., 2018a; Schwinger and Tjiputra, 2018; Jeltsch-Thömmes et al., 2020). Such experiments generally involve huge rates of negative emissions that are applied abruptly rather

than being phased in gradually. Also, the question whether pre-industrial climate can be re-established is interesting for our fundamental understanding of the climate system, but less so for current policy processes given that net negative emissions will certainly not be available in sufficient amounts to lower atmospheric $CO_2$ concentrations to pre-industrial values (e.g., Field and Mach, 2017). Some modelling studies have designed their CDR pathways based on more realistic assumptions

(Tokarska and Zickfeld, 2015; Tokarska et al., 2019; Li et al., 2020; Sanderson et al., 2017; Palter et al., 2018), and more recently a scenario including large amounts of net negative emissions has become available for use in the Coupled Model Intercomparison Project phase 6 (CMIP6, Eyring et al., 2016) ScenarioMIP (O'Neill et al., 2016), although this scenario is not suitable for reversibility studies since there is no reference pathway without overshoot.

While it is well established that $CO_2$ emissions that remain in the Earth system cause irreversible climate change (e.g., Solomon et al. 2009; Frölicher and Joos, 2010; Joos et al., 2011), the term "(ir)reversibility" of climate change has also been used to describe whether (or not) the climate system will, if $CO_2$ is removed from the atmosphere, return to a reference state where no CDR has been applied. However, there are several aspects of this usage of reversibility that are not well defined in the existing literature. First, the timescale of reversibility needs to be considered. Since there is considerable hysteresis in virtually all aspects of the climate system when CDR is applied, a variable that is irreversible on a short timescale might be reversible on a longer time horizon. Second, since there is internal climate variability, a condition as to when a variable is considered to have returned to the reference state is not trivial and needs to be defined. Third, there are different definitions of the reference state: (*i*) defined in terms of atmospheric $CO_2$ concentration (most often pre-industrial) that is restored in model simulations (e.g., Boucher et al., 2012; MacDougall, 2013), (*ii*) defined as a certain global mean temperature level (e.g., 1.5°C or 2°C above pre-industrial temperature) that is reached for pathways with and without CDR (Sanderson et al., 2017; Palter et al., 2018), or (*iii*) defined as a certain amount of cumulative carbon emissions that is reached with and without CDR (e.g., Tokarska and Zickfeld, 2015; Tokarska et al., 2019). All three definitions of a reference state have their specific advantages and drawbacks (for example, the definition based on a temperature target is preferable if the goal is to compare the impacts of an overshoot in reaching exactly that target) and different technical requirements (for example, the definition based on cumulative emissions requires that an Earth system model is run in emission-driven mode). From an Earth system perspective that includes all carbon cycle processes, the definition based on cumulative emissions is arguably the most natural, since it is anchored in the concept of proportionality between warming and cumulative carbon emissions (TRCE, transient climate response to cumulative carbon emissions) that has been shown to approximately hold over a wide range of emission pathways (e.g., Zickfeld et al., 2012; Krasting et al., 2014; Zickfeld et al., 2016). By using cumulative carbon emissions as the basic "currency", irreversibility emerges as a deviation from the paradigm that climate change is largely independent of the emission pathway and only depends on the amount of cumulative carbon emissions.

Studies that consider such a full Earth system perspective are still rare (Tokarska and Zickfeld, 2015; Tokarska et al., 2019; Li et al., 2020) and are all based on the same Earth system model of intermediate complexity. In this work, we complement these studies by investigating the reversibility of the Earth system after temperature overshoots of different magnitude and duration using a fully coupled state-of-the-art Earth system model (ESM) that includes a representation of permafrost carbon. Our overshoot simulations are idealized, but less idealized than in many of the previous studies cited above: our emission pathways have a period of increasing emissions, a period of decreasing emissions, and the positive and negative emission phases are smoothly joined. As a reference simulation, we take a pathway that stays well below 2°C warming without overshoot. Since, for this study, we are interested in simulating the response of the unperturbed Earth system to negative emissions, we do not

simulate a specific CDR method that would manipulate terrestrial or marine carbon sinks (e.g., bioenergy with CCS or ocean alkalinization). The time scale of reversibility considered here is 100 to 200 years after cessation of all emissions. We define

"reversibility" based on a reference pathway without overshoot (i.e., no CDR applied), and based on cumulative carbon emissions (i.e., the overshoot simulations have the same amount of cumulative carbon emissions after CDR than the reference pathway).

We note that our model, during phases of negative emissions, shows a pronounced surface temperature decline below the reference pathway at northern high latitudes that is related to the reduced northward heat transport by the Atlantic Meridional

Overturning Circulation (AMOC). Such cooling during the application of CDR seems to be a robust feature of ESMs that show a high sensitivity of AMOC to climate change (Schwinger et al., 2022). We refer the reader to Schwinger et al. (2022) for a detailed discussion of the temperature evolution during the negative emission phases in our model. Here, we focus on the longer term (i.e., after all emissions cease), and investigate the impact of overshoots on the terrestrial and marine carbon cycle as well as on large scale key indicators of Earth system (ir)reversibility.

**2 Methods**

**2.1 Model simulations**

We have conducted a set of seven idealized simulations using the Norwegian Earth system model version 2 (NorESM2-LM, Seland et al., 2020; Tjiputra et al., 2020). This set of simulations comprises one reference simulation where global warming levels reach approximately 1.7°C in the long term without overshoot and six simulations with different overshoot magnitude

and duration. Our simulations are emission driven, that is, atmospheric $CO_2$ concentrations evolve in response to $CO_2$ emissions and to atmosphere-ocean and atmosphere-land $CO_2$ exchanges. $CO_2$ emissions into the atmosphere is the only forcing applied. Land use and non-$CO_2$ greenhouse gas forcings are kept at pre-industrial levels. We note that NorESM2-LM has a transient climate response (TCR) of 1.48 K, which is at the low end of the CMIP6 model ensemble (Tokarska et al., 2020). Consistent with the relatively low TCR, NorESM2 also shows a low transient climate response to cumulative emissions

(TCRE) of 1.32°C EgC$^{-1}$ (CMIP6 range 1.32-2.30; Arora et al. 2020) and therefore shows relatively little warming for a given amount of $CO_2$ emissions.

NorESM2 employs version 5 of the Community Land Model (CLM5, Lawrence et al. 2019), which is capable of simulating key thermal, hydrologic, and biogeochemical processes associated with permafrost and their response to climate change. Compared to previous model versions, CLM5 includes several improvements (increased soil depth, improved vertical

resolution particularly in the top 3 m, vertically resolved soil biogeochemistry, and changes to modelled snow density, among others) enabling more realistic modelling of permafrost and active layer dynamics (Lawrence et al. 2019). The permafrost region is defined here as the geographic area where the model simulates a maximum active layer thickness shallower than 3 m. We note that CLM5 does not simulate the spatial dynamics of vegetation cover and competition between different plant functional types. For example, northward expansion of plant species due to climate warming is not represented. Changes in

carbon stocks associated to vegetation are therefore owed to changes in the plant carbon metabolism response to variations in atmospheric $CO_2$ and temperature only.

Our simulations are subdivided into phases of positive and negative emissions and include a period with zero emissions at the end. The positive emission trajectories follow the Zero Emissions Commitment Model Intercomparison Project (ZECMIP) protocol (Jones et al., 2019): Emissions are constructed as bell shaped curves with 50 years of increasing emissions and 50

135     years of decreasing emissions (see Fig. 2 of Jones et al. 2019, and Fig. 1a herein). Negative emission trajectories are constructed in the same way, but with a negative sign. The reference simulation, referred to as $B^{1500}$, has cumulative carbon emissions of 1500 Pg during the first 100 years, and zero emissions afterwards until year 400.

Each of the six overshoots is also simulated for 400 years in total. They are branched from simulations that follow the same emission profile as the reference $B^{1500}$, but with higher emissions (referred to as $B^{1750}$, $B^{2000}$, and $B^{2500}$, the superscript

indicates the amount of cumulative emissions, Table 1). The negative emission phases of the overshoots also last for 100 years, and we apply 250, 500 or 1000 Pg of CDR to simulate overshoots of different magnitude. To simulate overshoots with a different duration, there is a phase of zero emissions for 100 years between the positive and negative emission phases for three of the six overshoots. All simulations are extended by a phase of zero emissions until year 400. After negative emissions cease, the amount of cumulative carbon emissions is the same for the reference simulation and all overshoots (1500 Pg C, Fig. 1e).

We refer to the six overshoot simulations as $OS_y^x$, where the superscript indicates the cumulative amount of CDR, and the subscript indicates the duration of the zero-emission phase between positive and negative emissions. For example, $OS_0^{250}$ refers to an overshoot simulation with CDR of 250 Pg C, which starts immediately after positive emissions cease. In the following text we also refer to these overshoots as "low", "medium", and "high" (250, 500 and 1000 Pg CDR, respectively), and as "short" and "long" (0 and 100 years between positive and negative emission phases). The simulation design is summarized in

Table 1. For the reference simulation as well as for the "low" and "high" overshoot cases, we have run 3 ensemble members. Although 3 ensemble members is not enough to derive robust statistics, it provides an idea of the magnitude of interannual to multidecadal internal variability in our model under the applied forcing.

Our simulations are idealized and are not meant to represent a specific socio-economic scenario or to be realistic in terms of technical and socio-economic feasibility. Neither are the qualifiers "high", "medium", "low", "long", and "short" meant to

judge their realism or feasibility. The carbon dioxide removal from the atmosphere is also idealized and is not meant to represent a specific CDR method. There is no manipulation of terrestrial or marine carbon sinks that many proposed CDR methods would rely on (e.g., bioenergy with carbon capture and storage or ocean alkalinization). The closest analogue to the CDR applied in our experiments would be the method of direct air capture with carbon capture and storage (e.g., Fuhrman et al., 2021) assuming permanent and perfect storage (no leakage back to the atmosphere). To put the positive and negative

cumulative emissions of our simulations into perspective, we mention that 2500 Pg C cumulative (positive) emissions used for the "high" overshoots are similar to SSP5-8.5 emissions of 2622±144 Pg C (CMIP6 model mean and standard deviation, Liddicoat et al., 2021), and 1500 Pg C used for the reference simulation without overshoot corresponds roughly to SSP4-6.0

emissions (1420±98 Pg C), with the more widely known mitigation scenario SSP2-4.5 having somewhat lower emissions than this (1273±84 Pg C).

Scenarios that are consistent with reaching 1.5°C towards the end of the century, typically contain up to 330 Pg C negative emissions (IPPC, 2018) until 2100, although the uncertainties surrounding the feasibility of such scenarios are large (e.g., Fuss et al. 2018). Also, the quoted 330 Pg C are gross negative emissions, but, in realistic scenarios, part of the gross carbon dioxide removal is used to compensate for residual, difficult to mitigate positive emission. Thus, our "low" overshoot cases with 250 Pg of net negative carbon emissions are consistent with the amount of negative emissions applied in IPCC scenarios. Given

the longer time-horizon of our simulations compared to IPCC scenarios, we also consider our "medium" overshoot cases roughly consistent with the order of magnitude of CDR applied in such scenarios. The amount of negative emissions in our "high" overshoot cases (1000 Pg C) is certainly beyond a realistic range.

## 2.2 Reversibility

Our experiment design aims at simulating, in an idealized fashion, emission pathways that eventually meet the temperature

target mentioned in the Paris Agreement of keeping global warming to well below 2°C after a period of overshoot (with global warming >2°C), and we ask whether the state of the Earth system towards the end of these pathways is similar, or reversible, compared to the reference pathway without overshoot. Our definition of reversibility relies on cumulative carbon emissions: All six overshoot simulations reach, after CDR has been applied, the same amount of cumulative emission as the reference simulation (1500 Pg C). For surface temperature to be reversible, the close-to-linear dependence of global warming on

cumulative emissions, known as the transient climate response to cumulative carbon emissions (TCRE e.g., Gillett et al., 2013), needs to remain valid under negative emissions. As mentioned above, irreversibility emerges as a deviation from a linear dependence of climate change on cumulative carbon emissions in this approach.

Many previous studies on Earth system reversibility have been conducted with Earth system models of intermediate complexity, where internal variability is limited (e.g., Tokarska et al. 2019, Jeltsch-Thömmes et al. 2020, Lie et al. 2022). For

fully coupled Earth system model simulations we need a thorough definition of reversibility in the presence of internal variability. Similar to the concept of "Time of Emergence" (e.g., Keller et al., 2014), we consider an aspect of modelled climate reversible if the ensemble mean of the overshoot simulation returns to the reference pathway within the range of internal variability. Since our ensemble is too small (3 members) to derive this internal variability from the ensemble spread of single years, we define a measure of internal variability as follows: For a given point in time $t$ we calculate the variance of an annually

and spatially averaged time-series $x_i(t)$ for ensemble member $i$ ($i=1…3$) over 11-years, centred at point $t$, and define internal variability (IV) as the square root of the mean of the variances of the 3 ensemble members:

$$IV(t) = \sqrt{\frac{1}{3}\sum_{i=1}^{3}\frac{1}{11}\sum_{k=-5}^{5}(x_i(t_k) - \bar{x}_i)^2}$$

This definition is similar to defining internal variability as the standard deviation ($1\sigma$) of a time-series over a period of 33 years, except that we compensate for a shorter time-interval by using several ensemble members. The 11-year sliding window
used here is consistent with the fact that we present most of our results as moving averages with the same window length (11 years). Defining internal variability as one standard deviation ($1\sigma$) of interannual variations leads to relatively conservative threshold for reversibility, compared to some previous studies, which used a $2\sigma$ definition of internal variability (e.g., Keller et al., 2014). Decadal and longer-term variability, which can be larger than interannual variations for some variables, is partly removed in our approach by averaging over three ensemble members.

Reversibility depends on the time scale considered, since most Earth system variables show a considerable hysteresis behaviour when CDR is applied (e.g., Boucher et al. 2012, Jeltsch-Tömmes et al. 2020). In this study, we evaluate reversibility for a centennial time horizon after all emissions cease (i.e., after all simulations have reached cumulative carbon emissions of 1500 Pg). We define $REV_t$ as the reversibility after a time-span $t$ (in years) from this point in time. When comparing overshoots of different duration, there are two possible choices for assessing reversibility. First, we can compare $REV_t$ for fixed $t$, which are
different years of the simulations for the short and the long overshoots. For example, $REV_{95}$ is year 295 of the simulation for the short overshoot (emissions cease at year 200), but year 395 for the long overshoot (emissions cease at year 300). Alternatively, we can compare reversibility for a given year of the simulations, hence comparing reversibility at different times after emissions cease. For example, when calculating reversibility at the end of our simulations (based on 11-year means for the years 390 to 400), we actually compare $REV_{195}$ for the short overshoots to $REV_{95}$ for the long overshoots. In this study,
we focus on the second option, arguing that an advantage of applying CDR sooner rather than later would be that the Earth system has more time to recover from the consequences of an shorter overshoot. Hence, we assess reversibility at the end of our simulations, that is $REV_{195}$ for the short overshoots to $REV_{95}$ for the long overshoots.

## 3 Results and discussion

### 3.1 Surface air temperature for positive and zero emission phases

In the *B*-simulations, global average near surface temperature (SAT) peaks approximately at year 80 (i.e., 20 years before positive emissions cease) at 1.86, 2.20, 2.35, and 3.01°C above the pre-industrial level for the simulations with emissions of 1500, 1750, 2000, and 2500 Pg C (Fig. 1b, Table 2). After this peak, SAT slowly decreases by 0.39 to 0.66°C over the next 100-150 years, consistent with the negative zero emissions commitment (ZEC) values found for our model (MacDougall et al., 2020). ZEC is defined as the global mean surface temperature deviation from the point in time when emissions cease (Fig.
1c). Our simulations show a strongly variable ZEC, which is first negative (up to -0.4°C) and then shows positive values (up to 0.5°C) about 100 to 200 years after emissions cease (later for the stronger emission cases). The temperature decrease is mainly seen north of 40°N, since there is a strong reduction in the strength of the Atlantic meridional overturning circulation (AMOC, Fig. 1d) during the positive emission phases, which leads to a reduced northward heat transport by the ocean. During

the positive emission phases, the AMOC strength at 40°N decreases by 11 to 17 Sv or 48-76% in our model (Table 2) with a stronger reduction observed in simulations with higher emissions. After positive emissions cease, there is a gradual recovery of AMOC strength, which is slower for higher emissions. The recovery of AMOC leads to increasing surface temperatures and to a second SAT maximum at years 258, 280, 290, and 388 in the $B^{1500}$, $B^{1750}$, $B^{2000}$, and $B^{2500}$ simulations, respectively, which coincides well with the point in time when AMOC and northward heat transport in the Atlantic recover to almost pre-industrial strength (vertical lines in Fig 1c and d). For a more detailed discussion of the evolution of AMOC and high-latitude SAT in our simulations, we refer the reader to Schwinger et al. 2022. The final temperature for the reference simulation without overshoot is 1.70°C above pre-industrial level.

## 3.2 TCRE and reversibility of surface air temperature

We find that global average SAT is reversible in our simulations, according to our definition, in all overshoots except the high and long overshoot $OS_{100}^{1000}$ (Fig. 1f and g, Table 2). In all six overshoot simulations, global SAT decreases below the temperatures found in the reference simulation towards the end of the negative emission phases. After this cooling phase, SAT slowly recovers towards the reference temperatures. In our experiments, negative emissions are applied to a model state where AMOC and northward heat transport are strongly reduced compared to the reference case (Fig. 1h). As a consequence, SAT drops stronger than expected from the reduction in radiative forcing alone, particularly in the region north of 40°N. This aspect of our simulations is discussed in Schwinger et al. (2022), and we refer the reader to this study for more details. Here, we focus on the final state of the Earth system after overshoots of different magnitude and duration, that is, on the last 10 to 50 years of our overshoot simulations. We note that previous studies with intermediate complexity ESMs (MacDougall 2015; Jeltsch-Thömmes et al. 2020) have shown that hysteresis and irreversibility generally increase with increasing climate sensitivity. Therefore, owing its low climate sensitivity, NorESM2 most likely shows a relatively high degree of reversibility compared to higher sensitivity ESMs.

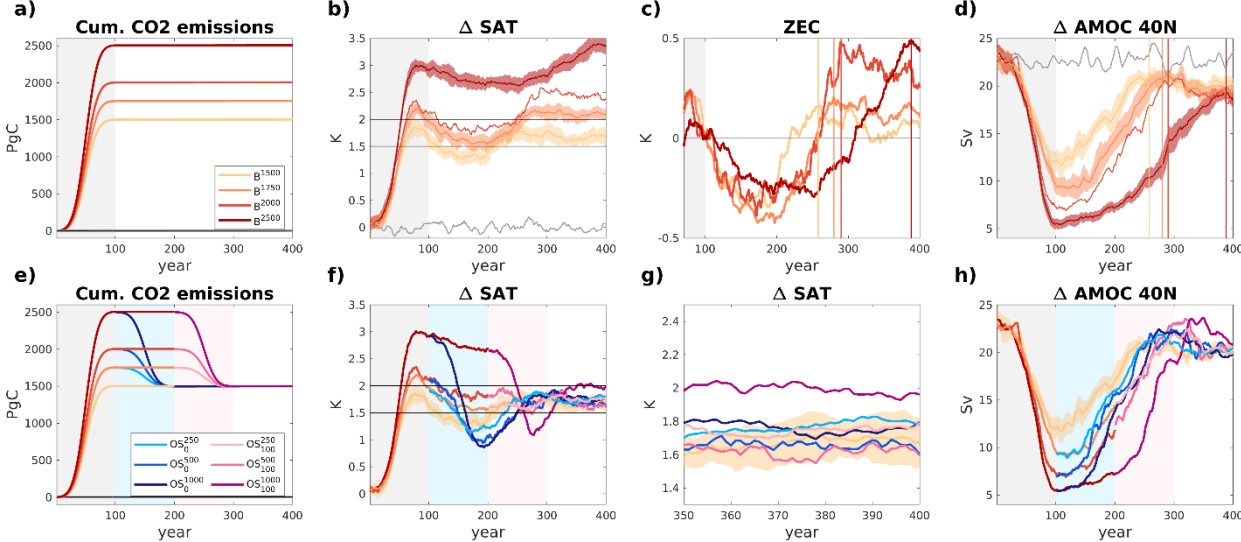

**Figure 1: Cumulative carbon emissions (a,e), global average surface temperature change (b,c,f,g), and change in AMOC strength (d,h). For clarity, panels a-d show ensemble means for the *B*-simulations only (as indicated in the legend in panel a) with positive emissions for 100 years and zero emissions thereafter, while panels e-h show ensemble means for the overshoot simulations branched from the *B*-simulations (as indicated in the legend in panel d). Panel c shows global average mean temperature change relative to**
250 **the point in time when emissions cease, and panel g provides a zoom into the last 50 years of the SAT evolution shown in panel f. The vertical lines in panels c and d indicate the time of the secondary SAT maximum, and the shading around the ensemble means indicates an estimate of internal variability as described in the text. An 11-year moving average filter has been applied to the time-series displayed in panels b-d and f-h. The shaded areas in the background indicate the different phases of the simulations (grey: positive emissions; light blue: negative emissions for the short overshoots; light red: negative emissions for the long overshoots).**

TCRE derived from our simulations shows a complex behaviour (Fig. 2). During the positive emission phases of the *B*-simulations, the relationship between ΔSAT and cumulative emissions is initially approximately linear and very similar across different rates of emissions. However, already before the peak in SAT (marked by stars in Fig. 2) is reached, there is a deviation from this linear relationship towards smaller values of TCRE. Calculated at the point in time when positive emissions cease (x-markers in Fig. 2), TCRE is smaller compared to the values at the SAT maxima by 0.13 to 0.25 K/EgC. Furthermore, during

the zero emission phases of the *B*-simulations, SAT decreases and recovers as described above. The final TCRE is between the values at the SAT maximum and at cessation of emissions, except for the highest emission simulation $B^{2500}$, where the final TCRE is substantially higher. We note however, that there is a decreasing temperature trend after the secondary SAT maximum in the $B^{2000}$ and $B^{2500}$ simulations. The secondary SAT maximum in $B^{2500}$ only occurs a few years before the end of our simulations, and therefore the high final TCRE value in $B^{2500}$ might be caused by the fact that this simulation is less

equilibrated at year 400 than the simulations with lower emissions.

We find that TCRE at the SAT maximum during the positive emission phases decreases with increasing emissions, i.e., TCRE is smaller by about 0.1 K/EgC for total emissions of 2500 Pg C compared to emissions of 1500 Pg C (stars in Fig. 2b). This behaviour is consistent with the study by Herrington and Zickfeld (2014). However, our findings are partly different from Krasting et al. (2014), where a decreasing trend was shown for cumulative emissions below approximately 2000 Pg C, and an

270 increasing TCRE trend was shown for simulation with larger emissions. The latter simulations, albeit, had rather low emission rates and might therefore not be comparable with our simulations. TCRE values calculated at the time when emissions cease in our simulations (x-markers in Fig. 2b) show no clear trend.

TCRE at the end of the $B^{1500}$ reference simulation and all overshoot simulations (Fig. 2c) reflect the final global mean near surface temperatures shown in Figure 1g (since cumulative emissions are the same for all overshoots and the reference). The
275 ensemble mean of each overshoot falls within the internal variability of TCRE found in the reference simulation except for the high and long $OS_{100}^{1000}$ overshoot. Within the groups of short and long overshoots there is a common pattern, where the medium size overshoot (500 Pg carbon removal) has the lowest final SAT and TCRE. Although there is only one ensemble member available for the medium sized overshoots, this result is consistent with other findings discussed in the next section.

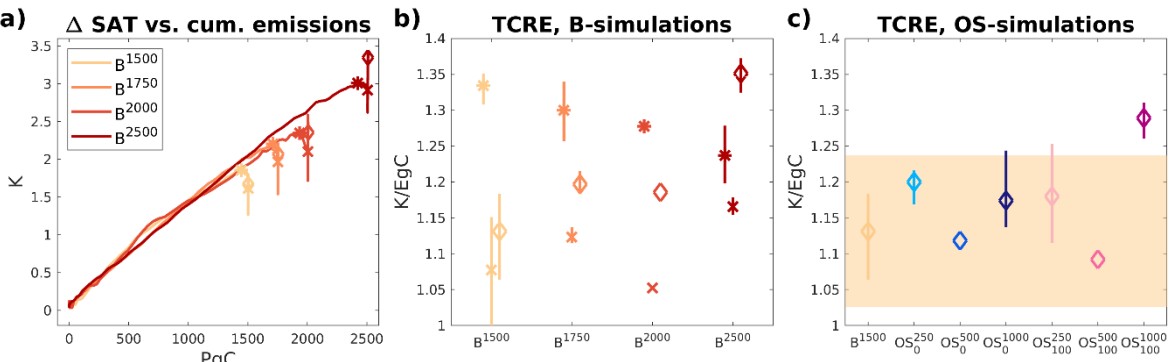

**Figure 2: Relationship between global mean SAT increase and cumulative carbon emissions (a), the value of TCRC at different points in time of the $B$-simulations (b), and TCRE at the end of the reference and the six overshoot simulations (c). For the $B$-simulations (panels a and b) the temperature maximum during the positive emission phases (stars), the temperature when emissions cease (x-markers), and the temperature at the end of the simulations (diamonds) are indicated. In panels b and c, the vertical bars**
**indicate the range of the three ensemble members and the marker symbol indicates the ensemble mean. The shaded area in panel c indicates the range of internal variability of the reference simulation $B^{1500}$. In all panels an 11-year running mean of SAT has been used.**

### 3.3 Global carbon cycle

Towards the end of our simulations, atmospheric $CO_2$ concentrations (Fig. 3a,b) converge to a similar level in all six overshoots
and the reference simulation, with differences less than 7 ppm (i.e., less than 15 Pg atmospheric carbon, Table 3) at year 400. According to our definition, however, the atmospheric $CO_2$ concentration is reversible only in the $OS_0^{250}$ and $OS_0^{1000}$ simulations, since the internal variability is small (about 1 ppm only). Nevertheless, given the vastly different pathways of atmospheric $CO_2$ concentrations in our simulations with freely evolving $CO_2$, a difference of less than 7 ppm is remarkable. We note that during the negative emission phases, atmospheric $CO_2$ concentrations decrease below the concentration in the
reference simulation in all overshoots. This effect is largest (about 50 ppm) for the large overshoots, which have the fastest rate of CDR.

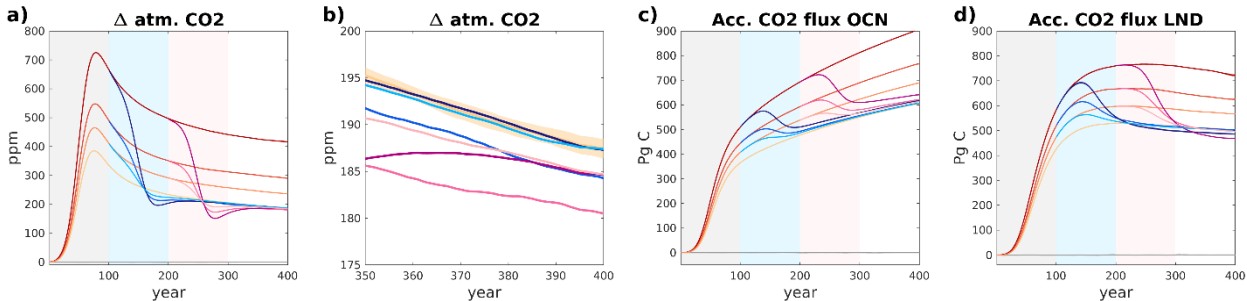

**Figure 3: Atmospheric CO₂ concentration (a,b), accumulated air-sea (c) and air-land (d) carbon fluxes in the *B*- and overshoot simulations. The color code for the simulations is given in Fig. 1. The shading around the ensemble means indicates an estimate of internal variability as described in the text. An 11-year moving average filter has been applied to all time-series. The shaded areas in the background indicate the different phases of the simulations (grey: positive emissions; light blue: negative emissions for the short overshoots; light red: negative emissions for the long overshoots).**

This similarity is caused by a near-perfect compensation between stronger ocean uptake and weaker land uptake during an overshoot (Fig. 3c,d), consistent with the findings of Tokarzka et al. (2019). The cumulative fractions of carbon stored in the atmosphere ($CF_A$), in the ocean ($CF_O$), and on land ($CF_L$) are shown in Fig. 4a to c. In all *B*-simulations, $CF_O$ increases continuously during the positive emission phase and the following phase of zero emissions, as the ocean circulation constantly removes carbon from the surface ocean to depth. $CF_O$ is lower for high (positive) emissions due to chemical and temperature mediated feedbacks (e.g., Arora et al., 2020). For land, the cumulative uptake fraction initially increases but shows a decreasing trend in the second half of the simulations, indicating that land becomes a source of carbon to the atmosphere. Same as for $CF_O$, and consistent with current understanding of carbon cycle feedbacks, $CF_L$ becomes smaller in the simulations with higher emissions.

During phases with negative emissions in the overshoot simulations, both land and ocean become a source of CO₂ such that their carbon stocks are reduced (Fig. 3c,d). For the ocean this happens as soon as the CO₂ partial pressure difference between atmosphere and ocean becomes negative. For the land, a reduced CO₂ fertilization effect shifts the overall balance between carbon uptake through net primary production and carbon release through heterotrophic respiration towards the latter. However, since these processes are slow and lag the reduction of the cumulative total of emissions, there is a rapid increase of $CF_O$ and $CF_L$ (Fig. 4b,c). At the end of the simulations, we find a common pattern between the group of short and the group of long overshoots: While $CF_A$ is almost equal in the low and high overshoots, it is lower by about 0.5 - 0.6% in the medium overshoots (Fig.4d), as reflected in the atmospheric CO₂ concentration towards the end of the simulation period (Fig. 3b). This is caused by a non-monotonic behaviour of the land carbon fluxes with increasing overshoot size. While the cumulative ocean uptake increases monotonically for larger overshoots (Fig. 4e), the land carbon fluxes show a more complex behaviour. Relative to the reference simulation, there is a decrease of land carbon stocks in all overshoots except for the $OS_0^{500}$ case where there is a small gain. Likewise, the long medium size overshoot $OS_{100}^{500}$ shows the smallest carbon loss among all long overshoot

simulations. Hence, somewhat counterintuitively, the overall land carbon storage is closest to the reference case in the medium sized overshoots (both long and short), which is caused by different levels of compensation between permafrost carbon losses and gains in other carbon reservoirs (vegetation and non-permafrost soils) as discussed further below in Sect. 3.5.

For the high overshoot cases, the compensating effect of land and ocean uptake is clearly exhibited in the cumulative fractions at the end of our simulations. For the $OS_{100}^{1000}$ simulation, $CF_O$ is about 2.5% (37.5 Pg C) larger, while $CF_L$ is 2.2% (33 Pg C) smaller than in the reference simulation, and these values show only small internal variability.

What impact on global average SAT could the small differences of up to 7 ppm in atmospheric $CO_2$ concentration due to the different emission pathways during the overshoots have? For a back-of-the-envelope calculation, we assume that the model

has already reached an equilibrium towards the end of the simulations, such that $\Delta SAT_{eq} = -F_{CO2}/\lambda$, where $\lambda=-1.34$ is the feedback factor for our model (Zelinka et al., 2020) and $F_{CO2}$ is the radiative forcing of $CO_2$ (Myhre et al., 1998). This relationship shows that a difference of 7 ppm atmospheric $CO_2$ concentration would result in an equilibrium SAT difference of about 0.06°C. Hence, this can potentially explain a part of the relatively low final TCRE value (Fig. 2c) in the $OS_0^{500}$ and $OS_{100}^{500}$ simulations compared to the other overshoots.

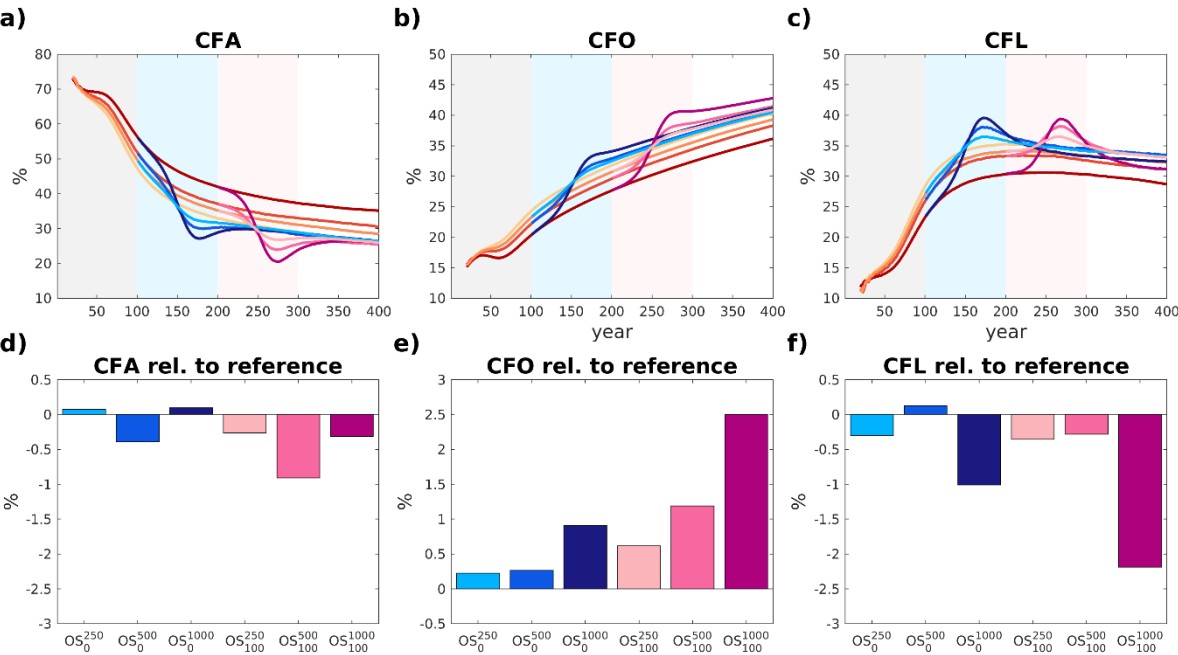

**Figure 4: Cumulative airborne (a), ocean (b), and land (c) fractions of emissions in percent of total emissions. Panels d-f show the deviation of the overshoots relative to the reference simulation without overshoot for the airborne (d), ocean (e), and land (f) fractions**
**at the end of the simulations (year 400). The color code for the simulations is given in Fig. 1. The shaded areas in the background of panels (a-c) indicate the different phases of the simulations (grey: positive emissions; light blue: negative emissions for the short overshoots; light red: negative emissions for the long overshoots).**

### 3.4 Terrestrial carbon cycle and permafrost

The northern hemisphere high-latitude permafrost extent follows the evolution of surface temperature over this area closely (Fig. 5). We define permafrost where the annual maximum active layer depth is shallower than 3 metres. During the positive emission phases, permafrost extent declines by between 5.80 to $9.65 \times 10^6$ km$^2$ from its pre-industrial value of $14.7 \times 10^6$ km$^2$. During the negative emission phases, northern high-latitude SAT cools temporarily below the reference level (Fig. 5c, see Schwinger et al. 2022 for details), and the permafrost area follows this pattern of warming-cooling-warming. Towards the end

of our simulations, permafrost area is reversible according to our definition (Fig. 5b, Table 3), consistent with previous studies, which show that the physical extent of permafrost area mainly follows the SAT trajectory and tends to recover under temperature reduction (Boucher et al. 2012; MacDougall 2013; Lee et al. 2019). It is, however, worth mentioning that landscape changes and hydrological responses to permafrost thaw, such as coastal erosion, excess ice melting, and formation of thermokarst lakes are highly heterogeneous and depend on small scale processes that are neither resolved nor parameterized

in our model. Therefore, irreversible changes at the (unresolved) landscape scale would occur even if the modelled large-scale physical state of the soil is found to be reversible according to our definition.

Vegetation carbon increases in all *B*-simulations during the positive emission phases due to the $CO_2$ fertilization effect (Fig. 6a-d). This effect remains dominant at high latitudes (north of approximately 45°N) throughout the 400 years of the *B*-simulations, while at lower latitudes the vegetation carbon stock declines again during the zero-emission phases. Vegetation

carbon is generally not reversible according to our definition, but this is mainly due to a low internal variability, and differences relative to the reference simulation remain small (below 8 Pg C globally, see further discussion below). Note that the distribution of vegetation is prescribed in our model, such that (potentially irreversible) changes in vegetation carbon as well as biophysical feedbacks (changes in albedo and roughness length) caused by shifts in vegetation composition are most likely underestimated in our simulations. For example, a northward tree-line expansion or shrubification in high latitudes due to

climate warming cannot be captured by our model.

Soil carbon stocks (Fig. 6e-h) increase during the *B*-simulations at all times and all latitudes, except in permafrost regions, where the release of carbon from thawing permafrost dominates. In non-permafrost regions, soil carbon stocks are generally larger after an overshoot compared to the reference pathway without overshoot at the end of the simulations (with the exception of the high latitude non-permafrost soil carbon in the $OS_0^{250}$ simulation, where soil carbon stock is slightly smaller). In the

permafrost region (Fig. 6h), the evolution of permafrost carbon stock is influenced by the temporary cooling during the negative emission phases. Due to the fact that SAT is colder in all overshoot simulations than in the reference pathway without overshoot for some period of time (Fig. 5c), the area affected by permafrost thaw is smaller for some period of time (approximately 100 years) than in the reference simulation (Fig. 5b). As a consequence, the loss of permafrost carbon is smaller in the $OS_0^{500}$ simulation than in the reference pathway $B^{1500}$. Also, the $OS_{100}^{250}$ and $OS_{100}^{500}$ simulations show a very similar

carbon loss that is only marginally larger than in the reference simulation.

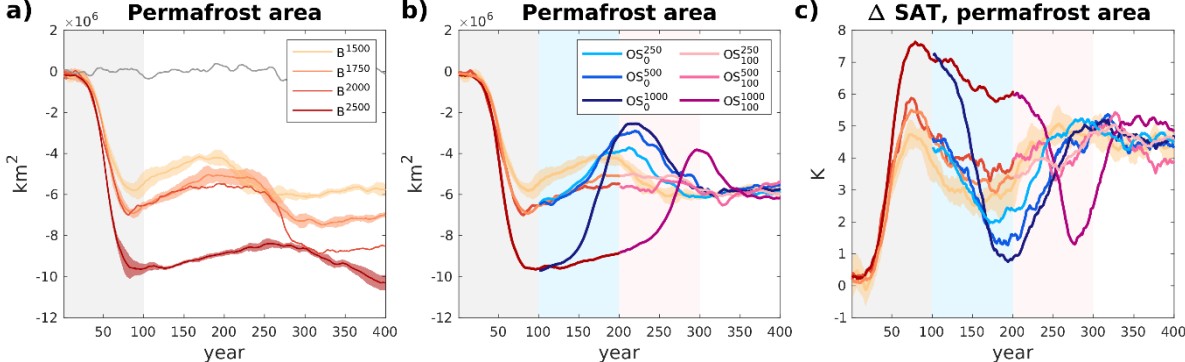

**Figure 5: Change in permafrost area relative to the pre-industrial control simulation. For clarity, panel a shows the *B*-simulations only with positive emissions for 100 years and zero emissions thereafter, while panel b show the overshoot simulations branched from the *B*-simulations. Panel c shows the surface temperature anomalies over the area that is occupied by permafrost in the pre-industrial control simulation. The shaded areas in the background indicate the different phases of the simulations (grey: positive emissions; light blue: negative emissions for the short overshoots; light red: negative emissions for the long overshoots).**

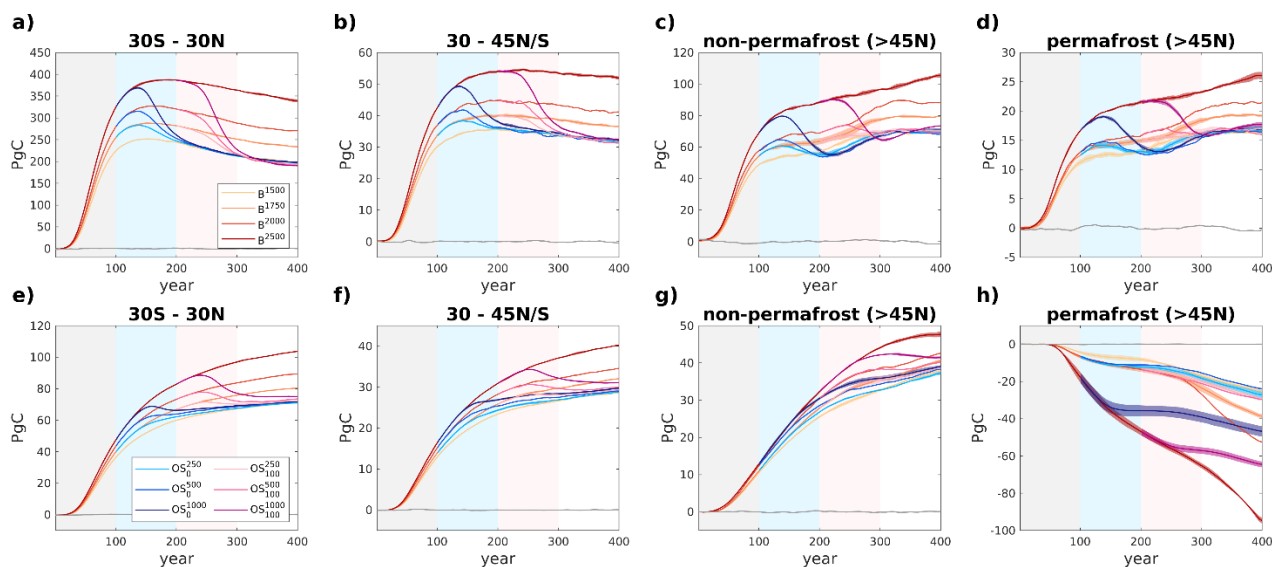

**Figure 6: Tropical (a), low latitude (30-45N, b), high latitude (>45°N) non-permafrost (c), and high latitude permafrost (d) vegetation carbon. Panels e-h show corresponding plots for soil carbon. The shaded areas in the background indicate the different phases of the simulations (grey: positive emissions; light blue: negative emissions for the short overshoots; light red: negative emissions for the long overshoots).**

### 3.5 Legacy of carbon stock changes after an overshoot

Figure 7 summarizes the changes in stock size of the main carbon reservoirs at the end of the six overshoot simulations (average over years 390-400) relative to the reference simulation $B^{1500}$ without overshoot. As pointed out above, there is a near perfect compensation between land and ocean carbon uptake in our model, that is, while land loses carbon relative to the reference simulation during an overshoot, the ocean gains additional carbon. This compensation leads to a very similar atmospheric $CO_2$ concentration in all overshoots compared to the reference simulation (Fig. 3). The main carbon reservoirs considered here are vegetation carbon, permafrost carbon, and non-permafrost soil carbon for land, as well as remineralised and preformed dissolved inorganic carbon (DIC) for the ocean. Changes in permafrost carbon are obtained as cumulative carbon fluxes summed over all permafrost grid cells, i.e. those grid cells that are defined as permafrost in the pre-industrial control simulation. Preformed DIC originates from atmospheric $CO_2$ that dissolves in the surface ocean and is transported into the interior by ocean circulation. In contrast, remineralized DIC has been transported into the interior ocean through the biological carbon pump: Biological uptake by planktonic organisms near the ocean surface, sinking to depth as particulate organic matter, and subsequent remineralization by bacterial activity. We note that the remineralization of organic carbon consumes oxygen (if present in sufficient quantity), such that oxic remineralization can be measured by apparent oxygen utilization (AOU), defined as the oxygen deficit in a water parcel relative to its saturated oxygen content.

Increased biological pump efficiency is the main driver of increased ocean carbon storage during and after an overshoot, which leads to a legacy of increased remineralized carbon in the ocean interior (see also Fig. 10). Changes in preformed dissolved inorganic carbon (preformed DIC) play a significant role only if the overshoot is long and intense (overshoots $OS_{100}^{500}$ and $OS_{100}^{1000}$), because in contrast to remineralized DIC, preformed DIC is lost from the ocean surface during the negative emissions phases of an overshoot when the partial pressure difference with the atmosphere becomes negative. Only during the long and high overshoots, there is sufficient time with a large positive $CO_2$ partial pressure difference to increase preformed DIC by significantly more than what is removed later during the negative emissions phases.

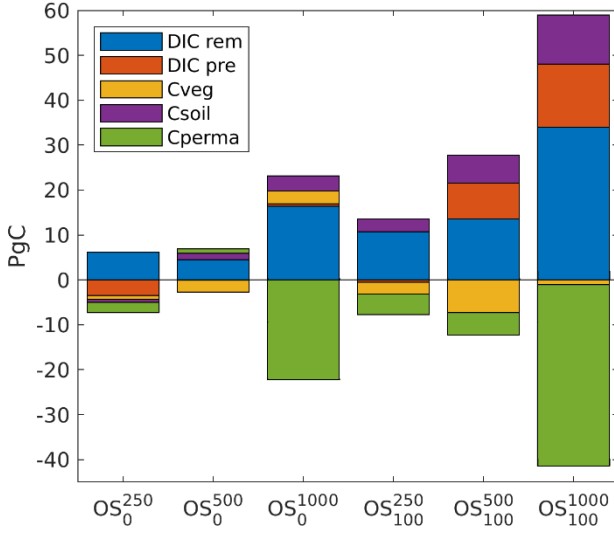

**Figure 7: Changes in stock size for the main carbon reservoirs of the Earth system at the end of the overshoot simulations (indicated along the x-axis) relative to the reference simulation without overshoot.**

Changes in soil carbon (permafrost and non-permafrost) dominate the legacy of the overshoots in terrestrial carbon stocks,

with permafrost carbon loss being far greater than non-permafrost soil carbon gain in the high overshoot simulations. We note that permafrost carbon is particularly sensitive to the large SAT fluctuations in northern high latitudes seen in our model (see Fig. 5c for SAT over the permafrost area). This is because permafrost is exclusively located in the area where the high-latitude cooling (north of 40°N) is large during the negative emission phases in our model. As mentioned above, during the negative emission phases, SAT drops below the reference level temperature for a significant amount of time, such that permafrost

carbon is actually better preserved during parts of the overshoots compared to the reference simulation with no overshoot. This is why the additional permafrost carbon loss remains relatively limited in the low and medium-size overshoots (there is even a small gain in the short medium-size overshoot). Permafrost carbon losses were larger if our model showed a weaker AMOC decline and associated high latitude cooling. In this case, the global budget of carbon stock changes could be dominated by losses from land. This could shift the overall carbon balance after an overshoot from lower to higher atmospheric $CO_2$

concentrations compared to a pathway without overshoot.

There are only relatively small differences in vegetation carbon after the overshoots compared to the reference simulation with no overshoot. This is to be expected since the vegetation carbon reservoir reacts with little time lags to changes in environmental conditions and $CO_2$. Generally, there is less carbon stored in vegetation after an overshoot, except for the $OS_0^{1000}$ simulation, which shows a small gain. We note that the inclusion of vegetation dynamics in our model would most likely affect

these results, since changes in biogeography would lead to larger changes in land carbon pools and larger time lags between drivers and response.

### 3.6 Marine heat uptake and sea level rise

Ocean heat content (OHC), like ocean carbon content, increases steadily during phases with positive or zero emissions (Fig. 8a). Unlike carbon content, however, the ocean heat content does not decrease significantly during phases of negative emissions (compare Figs. 3c and 8a; note there is a small decrease in OHC in the $OS_{100}^{1000}$ simulation around year 250). As a result, OHC after an overshoot deviates much more from the reference simulation with no overshoot, than the accumulated carbon fluxes. While the $CO_2$ partial pressure difference between atmosphere and ocean becomes negative during the negative emission phases (i.e., driving a carbon flux out of the ocean in the global mean), SAT does not decrease enough to cool the ocean significantly in the global mean. Also, the relatively strong AMOC reduction in our model increases the oceanic heat uptake by reducing heat losses to the atmosphere in the North Atlantic (Drijfhout, 2015).

These results imply that steric sea level rise (Fig. 8b, Table 3) is irreversible during the 400 years of all overshoot simulations, and higher than in the reference simulation by up to 15 cm for the most extreme overshoot $OS_{100}^{1000}$ (Fig. 8b, Table 3). This is consistent with a multitude of previous studies (e.g., Boucher et al. 2012; MacDougall 2013; Tokarska and Zickfeld 2015, Ehlert and Zickfeld 2018), which show that sea-level rise is largely irreversible on centennial to millennial time scales. We note, however, that negative emissions are indeed effective in reducing the rate of sea level rise after an overshoot to a value similar or lower than that of the reference simulation (Fig. 8c). The additional steric sea level rise due to an overshoot remains relatively small compared to the sea level rise committed to in the reference simulation in our model (<20% at year 400 except for the most extreme overshoot $OS_{100}^{1000}$). Thus, the rate of sea level rise determines the pace and cost of necessary adaptation for the decades to centuries after an overshoot. Therefore, limiting the rate of sea level rise after an overshoot might arguably be more policy relevant in the context of negative emissions than a relatively limited contribution of the overshoot to the sea level rise itself.

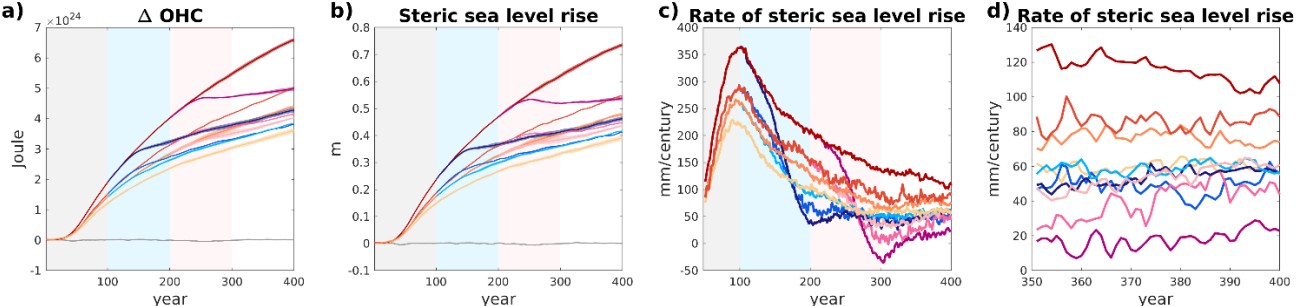

**Figure 8: Ocean heat content (a), steric sea level rise (b), and rate of steric sea level rise smoothed over 50 years (c,d). Panel (d) provides a zoom into the last 50 years of panel c. The color code for the simulations is given in Fig. 1. The shading around the ensemble means indicates an estimate of internal variability as described in the text. An 11-year moving average filter has been applied to the time-series in panels a and b. The shaded areas in the background indicate the different phases of the simulations (grey: positive emissions; light blue: negative emissions for the short overshoots; light red: negative emissions for the long overshoots).**

### 3.7 Stressors for marine ecosystems

The evolution of global average marine net primary production (PP) in the *B*- and overshoot simulations is shown in Fig. 9a and b. During the positive emission phases, PP decreases by between 2.0 and 2.8 Pg C yr$^{-1}$ from its pre-industrial value of 33.6 Pg C yr$^{-1}$. During the zero emission phases of the *B*-simulations, there is a gradual recovery of PP except for the highest emission simulation $B^{2500}$. In general, the recovery of PP is weak compared with, for example, the recovery of AMOC strength (Fig. 1d). Global PP is reversible according to our definition for all overshoot simulations (Fig. 9b, Table 3). We note that in our model the export of particulate organic carbon (POC, not shown) from the surface ocean shows the same qualitative behaviour as PP. Consistent with previous studies (Schwinger et al., 2014; Arora et al., 2020), the reduced PP and carbon export is overcompensated by a reduction of ocean circulation and upwelling of nutrients and carbon from the deep ocean under climate change, such that the remineralized component of DIC and nutrients in the interior ocean increases steadily in all simulations. This process is irreversible at the 400-year timescale of our simulations such that the excess of remineralized phosphate over the pre-industrial values is larger by up to 30% in the overshoot simulations compared to the reference simulation (Fig. 9c).

The global average surface oxygen concentration (not shown) is mostly driven by surface temperature and is, consistent with our results for SAT, reversible for all but the $OS_{100}^{1000}$ overshoot. Surface pH (not shown) is not formally reversible according to our definition with the exception of the short and low overshoot $OS_0^{250}$. This is, however, due to the very low internal variability of global mean surface pH, and we note that surface pH is actually higher (i.e., slightly closer to pre-industrial values) towards the end of all overshoot simulations. This behaviour has been observed in a previous study (Li et al. 2020) and can be explained by the fact that atmospheric $CO_2$ has been lower than in the reference simulation for a period of time before the reference level of atmospheric $CO_2$ is approached (Fig. 3a).

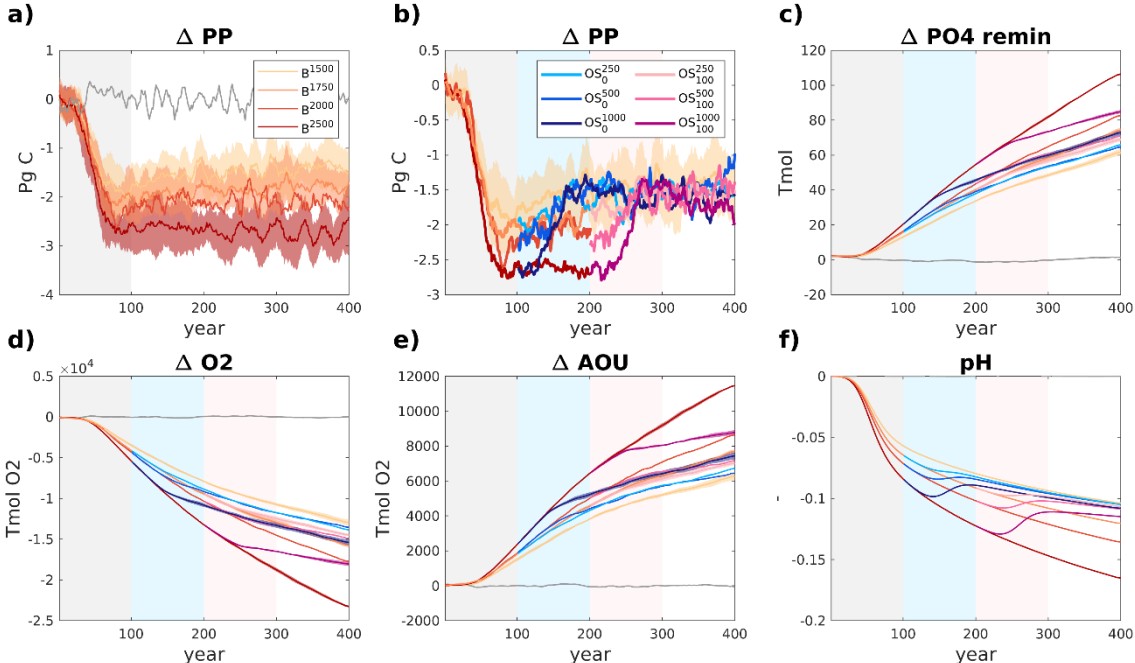

**Figure 9: Change in marine primary production (a,b), change in remineralized phosphate (c), change in global ocean oxygen content (d), change in global apparent oxygen utilization (e), and change in global average pH (f). For clarity, panel a only shows the $B$-simulations, and panel b only shows overshoot pathways. The shading around ensemble means indicates an estimate of internal variability as described in the text. An 11-year moving average filter has been applied to all time-series. The shaded areas in the background indicate the different phases of the simulations (grey: positive emissions; light blue: negative emissions for the short overshoots; light red: negative emissions for the long overshoots).**

Global ocean oxygen content is not reversible in any of the overshoot simulations (Fig. 9d, Table 3). The non-reversibility in oxygen content is governed in approximately equal parts by ocean warming and changes in the biological pump in our model. This can be seen from Apparent Oxygen Utilization (AOU, Fig. 9e), which is roughly 2500 Tmol larger at the end of the $OS_{100}^{1000}$ overshoot than in the reference simulation, explaining about half of the 5000 Tmol non-reversibility of global oxygen content. The other half can be attributed to ocean warming, which reduces the solubility of oxygen and thus the interior oxygen content once surface waters are transported into the interior ocean.

Global average pH Fig. 9f) is not reversible either, although the differences relative to the reference simulation are small except for the most extreme overshoot $OS_{100}^{1000}$. That is, the negative emissions are generally effective in bringing pH close to the reference simulations, and the non-reversibility (according to our definition) is due to the very small internal variability of global average pH.

To understand the spatial structure of interior changes of oxygen, remineralized DIC (and phosphate), as well as pH, it is instructive to look at changes of the simulated ideal age of water masses (Fig. 10a-c for the long overshoots in the Pacific, see

Figs. A1-A3 for corresponding figures for the Atlantic basin and the short overshoots). In the reference simulation $B^{1500}$, the
simulated ideal age relative to the pre-industrial state increases by more than 100-200 years everywhere below approximately
2000 m depth due to increasing ocean stratification. Consequently, the water masses that are upwelled in the Southern Ocean
and reach the surface around 60°S to 70°S are getting older, too. Above these ageing water masses, there is a region where
water mass ages get younger under ongoing climate change. The core of this region extends from approximately 30°S to 30°N
and the increased ventilation here is caused by a reduction in vertical transport of older waters from below (Gnanadesikan et
al., 2007), a pattern that is commonly found in ESMs (Cabré et al., 2015). During the overshoot simulations these patterns are
amplified, such that most of the old waters in the deep ocean get older while the ventilation of intermediate water masses
generally increases.

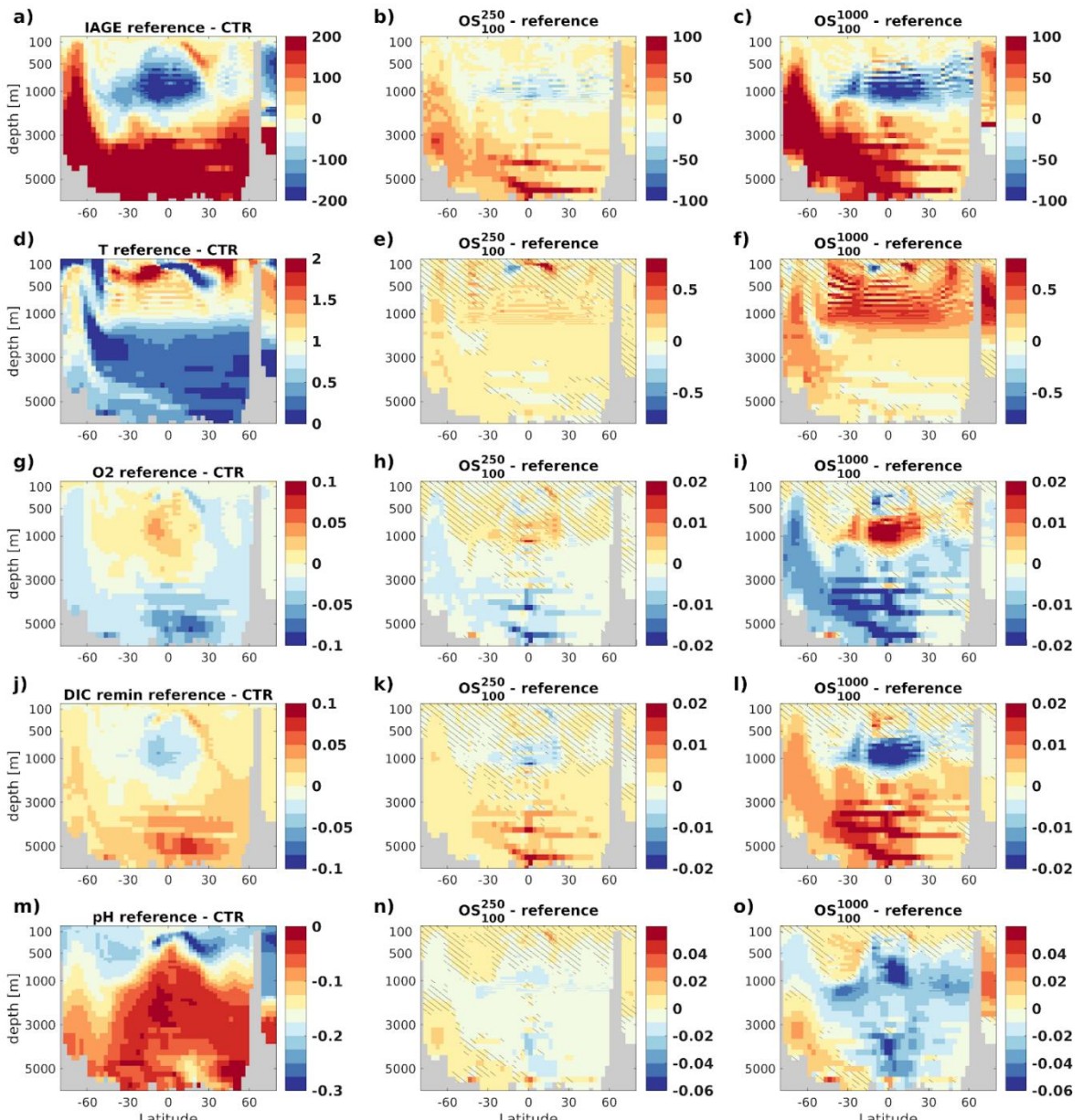

Figure 10: Anomalies of sea water ideal age (a-c), potential temperature (d-f), dissolved oxygen (g-i), remineralised DIC (j-l), and pH (m-o) for a zonal mean section through the Pacific (between 180°W and 140°W). The first column of panels displays the difference between the reference simulation $B^{1500}$ and the pre-industrial control run (CTR), while the second and third columns of panels show the anomalies of the long overshoot simulations $OS^{250}_{100}$, and $OS^{1000}_{100}$ relative to the reference. Shown is the ensemble mean averaged over the last eleven years of our simulations (years 390-400), and the hatched areas indicate reversibility (REV₉₅) according to our definition. Corresponding figures for a section through the Atlantic Ocean and for the short overshoots are shown in Appendix A.

The influence of sea water temperature is mostly limited to the upper 1000 to 2000 metres in the Pacific but reaches down to about 3000 metres with Atlantic deep waters (Fig. A1). Temperature changes in the upper ocean (surface down to 500-1000 metres) are largely reversible, although, for the most extreme overshoot $OS_{100}^{1000}$, the volume of water with reversible temperature changes becomes small, particularly in the Atlantic Ocean. Below 1000-2000 meters depth temperature changes (but also changes in the other variables discussed here) are largely irreversible. This is partly because the internal variability in the deep ocean is small, and this favours the emergence of non-reversibility according to our definition. However, for changes mediated by the biological pump (oxygen, remineralized DIC) this is not necessarily true. Here, non-reversible changes in the deep ocean can be as large or even larger than in the upper ocean (Fig. 10g-l). Expectedly, the changes in interior oxygen content (Fig. 10g-i) show a negative correlation with the changes in ideal age, as do the changes in the remineralized component of DIC (Fig. 10j-l) and nutrients (not shown). Note that remineralized DIC is based on a preformed phosphate tracer in our model, such that it is not derived from oxygen via apparent oxygen consumption. Where water masses get older, oxygen content decreases, because more oxygen is used for remineralization of organic matter. Consequently, in the Pacific, the loss of oxygen is largest in the tropical deep ocean and in the Southern Ocean upwelling, where also the legacy of the overshoots is largest (Fig. 10g-l).

Changes in pH (Fig. 7m-o) mainly depend on the surface history of a water parcel and ventilation pathways, although changes in organic matter export and remineralization as well as changes in $CaCO_3$ export and redissolution contribute to pH changes at depth (Lauvset et al., 2020). Close to the surface, pH is mostly reversible, since surface pH follows atmospheric $CO_2$ concentrations closely. At locations in the surface ocean where pH is not reversible, the pH value is actually higher than in the reference simulation, reflecting the fact that atmospheric $CO_2$ has been lower during the negative emission phases of the overshoots for some time (Fig. 3a). Likewise, in the deep-water formation regions of the Southern Ocean, we find pH values that are higher after an overshoot. At intermediate depth, there is a legacy of lower pH values in water masses that have ventilated these depths during the overshoot periods. In the deep ocean, mainly the Pacific, we find irreversibly lower pH values that are most likely caused by an increase in remineralized carbon (these water masses have not been ventilated during the course of the overshoot simulations). We note that our model simulates too large marine primary production in the equatorial Pacific and excessive oxygen minimum zones below, such that the effect of changes in remineralized carbon in the deep Pacific on pH might be overestimated.

So far, we have chosen to assess reversibility for the same simulation year, that is, we compare REV$_{95}$ for the long overshoot simulations (95 years after negative emissions ceased, Fig. 10) to REV$_{195}$ for the short overshoot simulations (195 years after negative emissions cease, Figs. A2 and A3). To allow for a clean comparison of the effect of overshoot length on reversibility, Fig. A4 shows REV$_{95}$ for the short overshoots (i.e., reversibility derived from simulation years 290-300). Comparing Fig. A4 and Fig. 10 reveals that the volume of water masses showing irreversible change tends to be larger in the long overshoots, although the spatial patterns are broadly similar. The main difference between the short and long overshoots is that the irreversible changes in the deeper ocean are much more pronounced for the long overshoot duration, indicating a clear benefit of limiting the duration of an overshoot.

### 3.8 Tipping points

It has been suggested that tipping points exist in the Earth system, which would, once a critical threshold is crossed, trigger self-amplifying feedbacks and "tipping cascades" that could lead to irreversible changes and a "hothouse Earth" climate state (Steffen et al., 2018; Lenton et al., 2019). Such tipping mechanisms could exacerbate the risk of relying on CDR to return to a less dangerous climate state after an overshoot. Our Earth system model simulations represent some of the proposed cascading tipping mechanisms, while others are not modelled. The release of carbon from permafrost soils is one of the sources that could increase atmospheric $CO_2$ concentrations and thus re-inforce global warming. In our simulations, permafrost carbon release is indeed the largest irreversible contribution to carbon losses from land during most of the overshoot simulations. However, continuous and century-long losses from land carbon stocks are compensated by increased ocean carbon uptake in our model, such that the atmospheric $CO_2$ concentration is not larger after an overshoot compared to the scenario with no overshoot.

Compared to other ESMs, our model simulates a relatively strong reduction, followed by a gradual recovery, of AMOC strength (Schwinger et al. 2022). Therefore, tipping points and cascades related to a strong weakening of AMOC should be represented in our simulations. We note that, due to the northern high-latitude multi decadal cooling trend simulated after emissions cease and the amplification of this cooling during phases of CDR, the AMOC slowdown actually helps to limit the impact of an overshoot on Arctic sea ice, permafrost and presumably the Greenland ice sheet (the latter is not represented in our simulations). We also do not see boreal forest dieback, even in the most extreme overshoot simulation considered here. In the high latitudes, vegetation carbon increases with increasing temperature in our model, and tends to be slightly higher after an overshoot compared to the reference simulation. For the Amazon rainforest, although vegetation carbon tends to be slightly lower after an overshoot, the system is stable, also after the most extreme overshoot, and there are no abrupt shifts. Although the ocean primary productivity decreases with progressing climate change, the efficiency of the biological pump (measured as the stock of remineralized carbon in the ocean) increases in our simulations, consistent with results from CMIP5 and CMIP6 ESMs (Schwinger et al. 2014, Arora et al. 2020). This increase is due to an overcompensation of reduced PP by reduced ocean circulation and upwelling. In general, we do not find abrupt large-scale shifts that would be indicative of tipping points. This result is broadly consistent with previous model studies using intermediate complexity ESMs (e.g., Steinacher and Joos 2016, Jeltsch-Thömmes et al. 2020) or CMIP6 ESMs (e.g., Koven et al. 2022). We have, however, not screened our model results for abrupt regional shifts as done, for example, in Drijfhout et al., 2015.

### 4 Summary and conclusions

We have simulated idealized scenarios that reach the goal of holding global average temperature increase to well below 2°C after a period of temperature overshoot with a state-of-the-art Earth system model. To assess whether climate change can be

partially reversed by CDR, we compare six overshoots of different magnitude and duration to a reference scenario without overshoot. We define an aspect of the Earth system to be reversible through application of CDR if the mean state after an

overshoot is within the internal variability of the reference case without overshoot. We stress that this definition neither implies reversibility in the absence of CDR nor reversibility of climate change that is committed to in the reference scenario. We also note that our Earth system model (NorESM2) has a low climate sensitivity, and that previous studies with intermediate complexity ESMs (MacDougall 2015; Jeltsch-Thömmes et al. 2020) have shown that hysteresis and irreversibility are generally smaller for low climate sensitivity.

In our overshoot simulations, atmospheric $CO_2$ concentrations return to the same or slightly lower levels than in the reference simulation, since a larger loss of land carbon during the overshoot period is (over)compensated by stronger ocean carbon uptake. Losses from land are mainly due to carbon release from thawing permafrost and a reduction of vegetation carbon stocks, while (non-permafrost) soil carbon stocks are generally larger after an overshoot. In the ocean, the legacy of the

overshoots is mainly seen as an increased stock of remineralized carbon, since reduced primary production during the overshoot periods is overcompensated by reduced upwelling and increased stratification (hence, the biological pump efficiency increases).

We further find that, on a timescale of 100 to 200 years after all emissions cease, CDR is effective in partially reversing the global mean state of many aspects of the Earth system (near surface air temperature, marine primary productivity, terrestrial net biome production, surface $O_2$ and pH, AMOC strength, permafrost extent), except in the most extreme overshoot scenario

with large emissions (comparable to SSP5-8.5), and a long period of time (100 years) before large $CO_2$ removals (1000 Pg C) are applied. Non-reversibility is generally found in the deeper ocean (below approximately 500 m depth), where sea water temperature, oxygen content, and pH all show a considerable volume where they are non-reversible even for the overshoots of lower intensity.

We do not find evidence for large-scale self-amplifying feedbacks or abrupt shifts that would indicate that a tipping point has been crossed during the overshoot periods in our model simulations. Losses of land carbon are compensated by increased ocean uptake, and the ocean biological pump efficiency rather increases than decreases during the wide range of overshoots considered here. There is no sign of tropical or boreal forest dieback in our simulations. Hence, we do not find evidence for most of the mechanisms invoked by Steffen et al. (2018), which would put the Earth system onto a trajectory towards a

"hothouse Earth" state, at least not during any of our overshoot simulations with our model.

There are, however, certain limitations in our simulations that hamper our ability to assess abrupt shifts or tipping points. First, the representation of ecosystems is simplified. For example, there is no dynamic response of vegetation to climate change in our model (i.e., no changes in biogeography), and effects like a northward tree-line expansion in high latitudes are not captured. Key marine ecosystems like coral reefs are not modelled at all and there is no representation of higher trophic levels. Further,

ice sheets are not represented in our ESM such that we cannot assess the impact of overshoots on ice sheet melt and associated sea-level rise as well as changes in ocean circulation. Further, although our model simulates carbon release through gradual

permafrost thaw, effects of abrupt permafrost thawing and related greenhouse gas emissions (Turetsky et al., 2020) are not included. Also, since methane emissions are not interactively coupled to the atmosphere, methane release from thawing permafrost is not considered in our simulations. Hence, the overall carbon release from permafrost soils is most likely underestimated. Likewise, methane emissions from wetlands, which have been found to react strongly to warming (Kleinen et al. 2020), do not affect atmospheric methane concentrations in our model configuration. Finally, we note that our idealized scenarios do not contain any other forcings than $CO_2$ emissions, and that land use stays at its pre-industrial state. Massive land use changes such as deforestation could precondition forests to be more prone to irreversible changes (Li et al., 2022).

The amount of carbon dioxide removal considered in this study, particularly in the medium and high overshoots (500 and 1000 Pg carbon removal, respectively), is most likely beyond a feasible range. The simulations that might be realistic in terms of feasibility of CDR, i.e., the low overshoots with 250 Pg carbon removal, show a large degree of reversibility. The non-reversible variables in these simulations (sea water temperature, oxygen, and pH in the deep ocean) have relatively small deviations from the reference state without overshoot ("small" compared to the changes that have been committed to by accepting a warming of well below 2°C as safe). Although the consequences of this non-reversibility in terms of impacts on ecosystems are uncertain, it seems that reversibility might not be the main concern when considering realistic overshoot scenarios (which might even have less CDR than the small overshoots with 250 Pg carbon removal considered here). Rather, the climate change impacts during the period of overshoot (which have not been assessed in this study) might be a factor that sets tighter limits to overshoot strategies than Earth system irreversibility.

We conclude with a few recommendations for further research. First, this is a single model study, and to estimate uncertainties related to realistic overshoot pathways, experiments that allow to assess reversibility should be included in future phases of model intercomparisons like CMIP7. Second, a more realistic representation of processes that have already been identified as main sources of irreversibility, for example carbon and methane release from permafrost thaw should be implemented into Earth system models with priority. Third, climate change impacts on ecosystems, including irreversible shifts and abrupt changes, are only poorly represented in Earth system models, but might arguably be the most critical risks of delayed mitigation and subsequent carbon dioxide removal. A better assessment of these risks in the context of overshoot trajectories is urgently needed.

## Appendix A

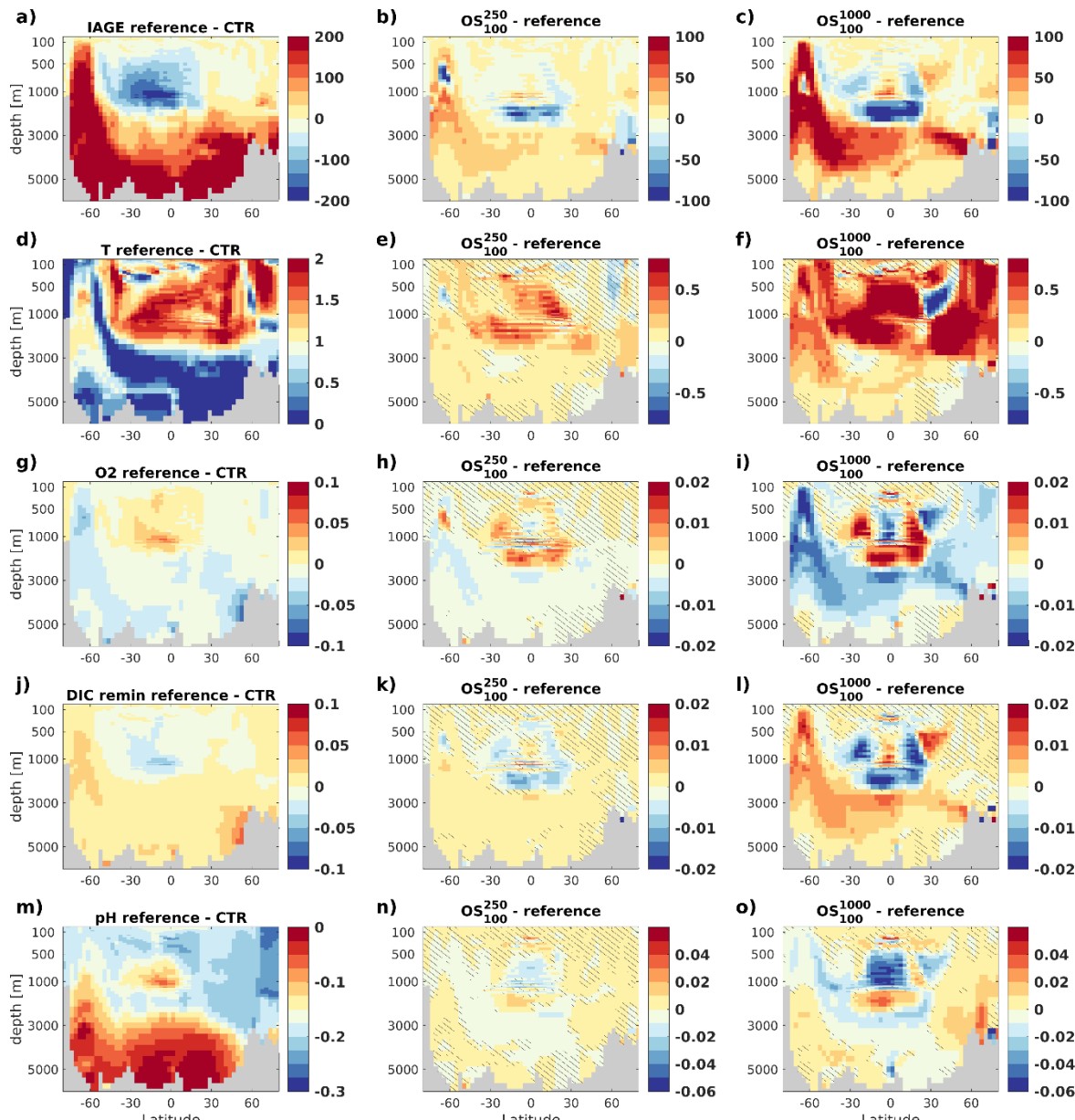

**Figure A1: As Figure 10, but for a zonal mean section through the Atlantic.**

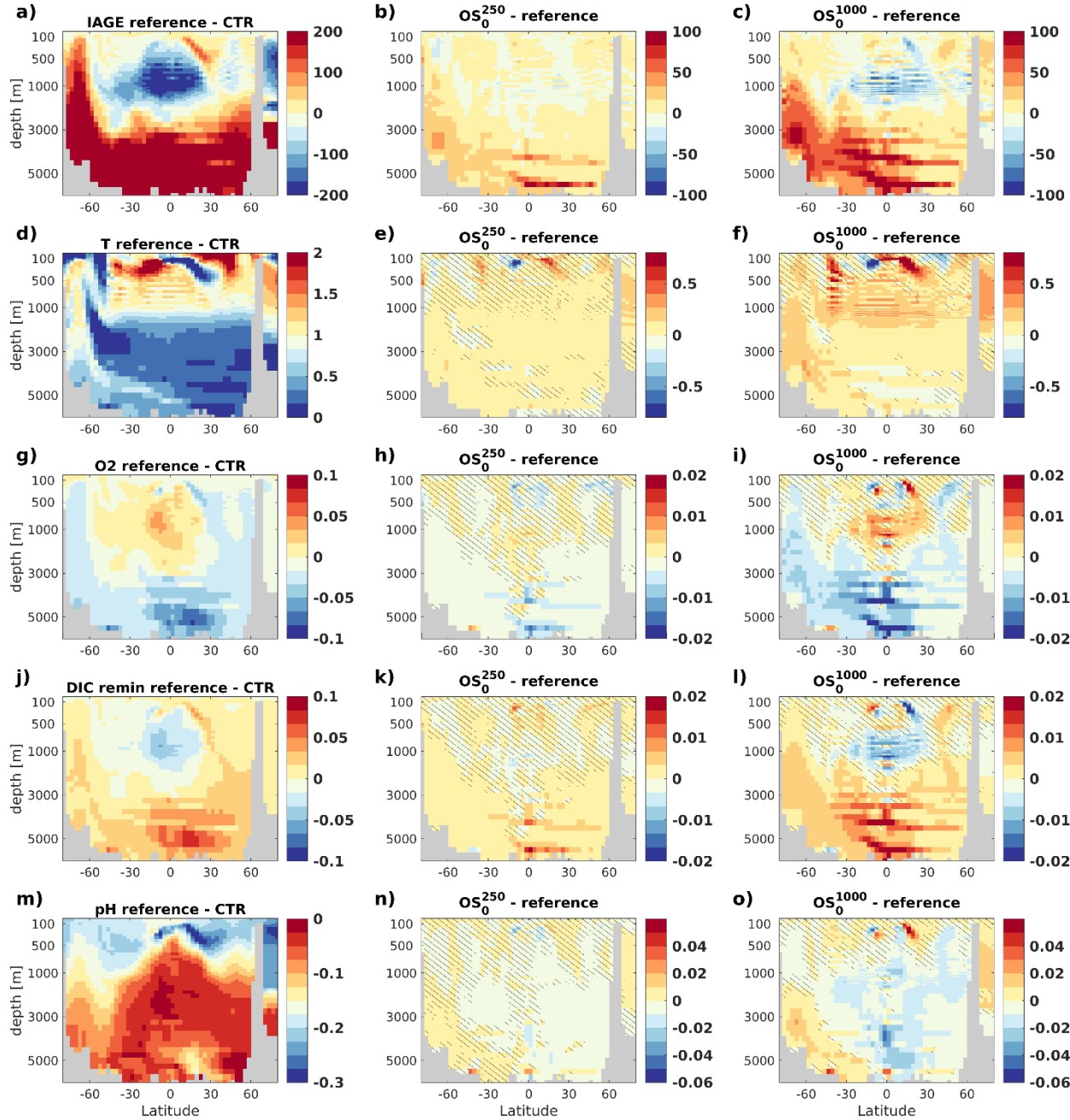

**Figure A2: As Figure 10, but showing REV195 for the short overshoot duration.**

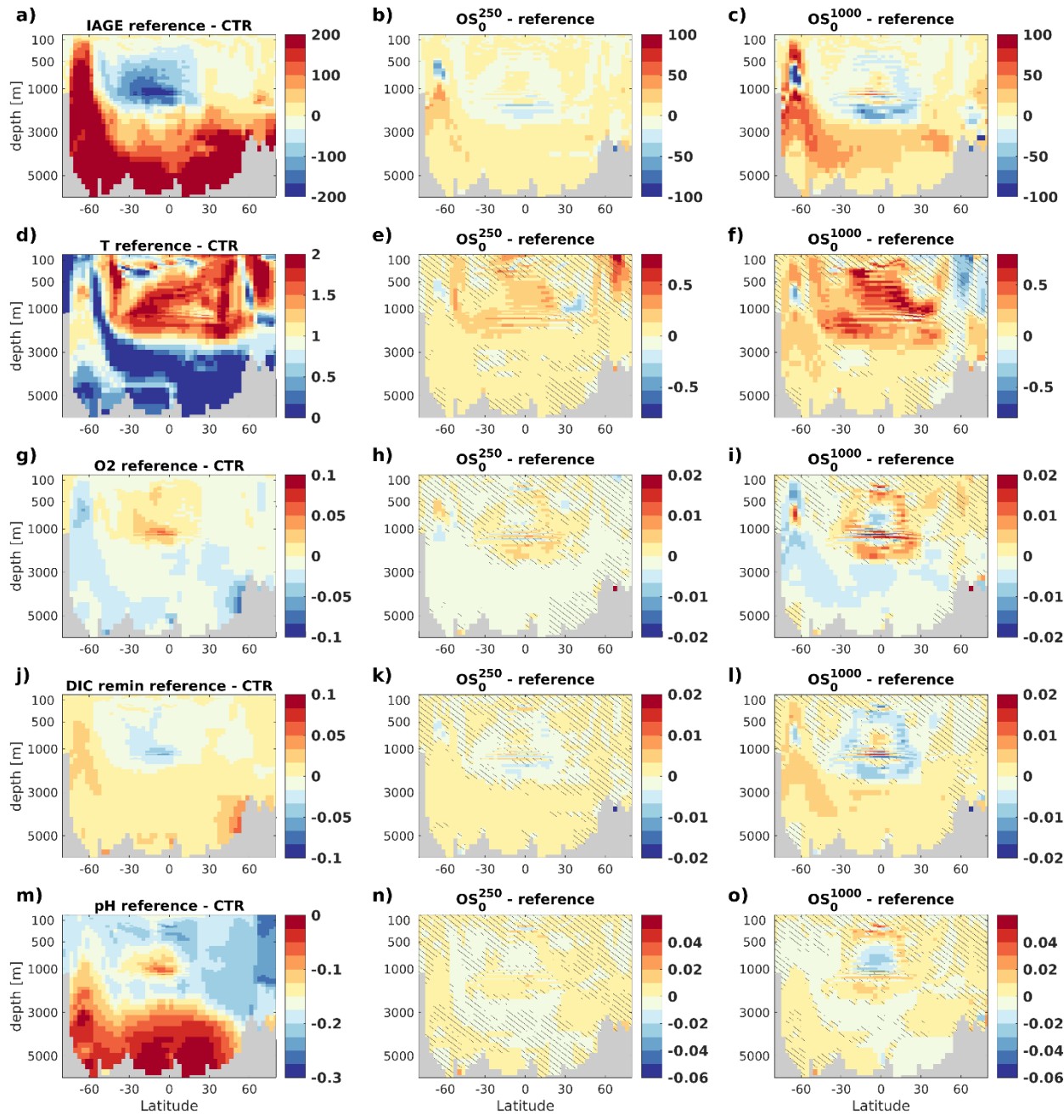

**Figure A3: As Figure 10, but showing REV<sub>195</sub> for a zonal mean section through the Atlantic and for the short overshoot duration**

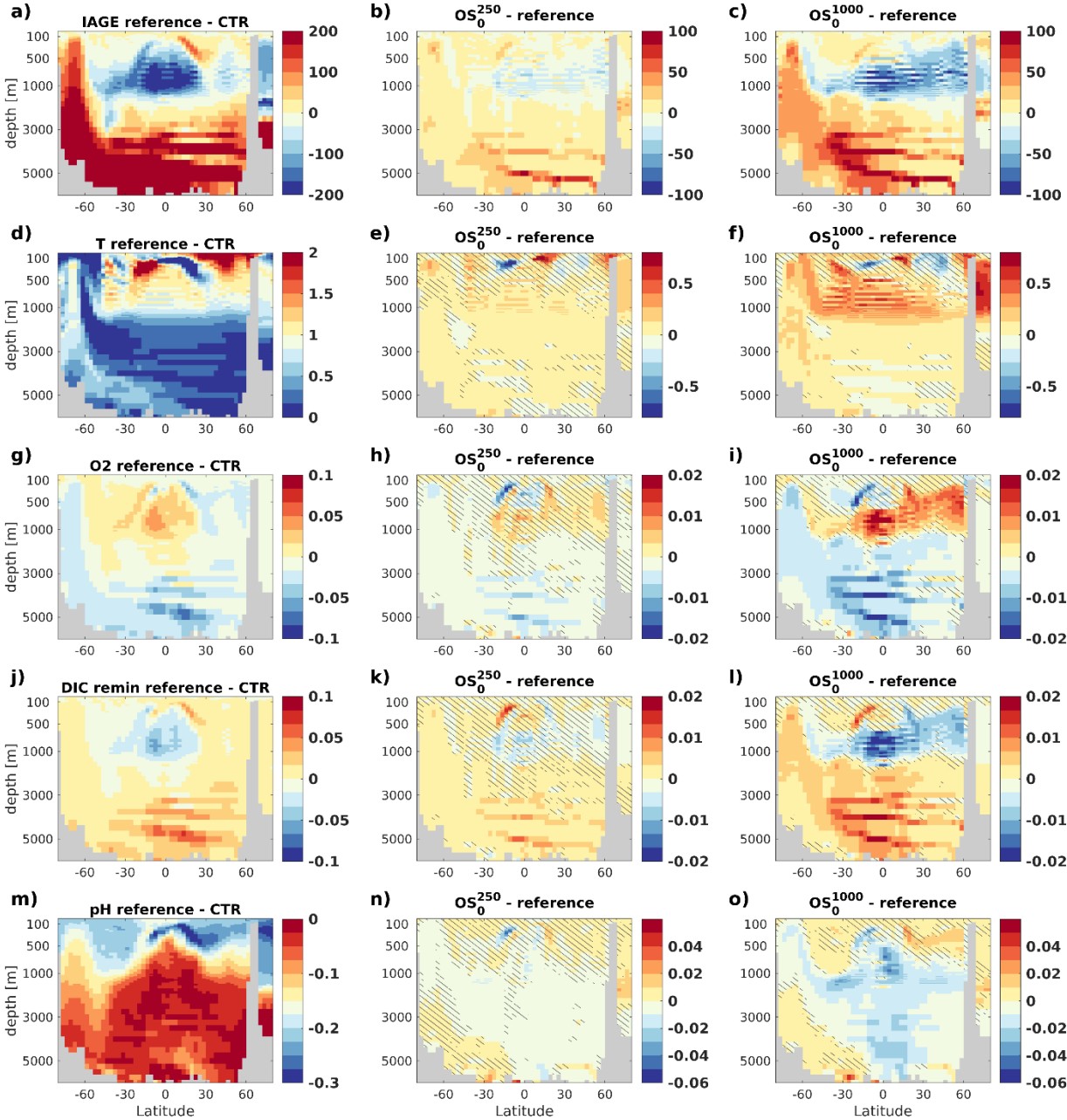

**Figure A4: As Figure 10, but showing REV$_{95}$ for the short overshoot duration for a zonal mean section through the Pacific**

**Code and data availability**

The source code of NorESM2 is available at https://doi.org/10.5281/zenodo.3905091. The model data generated in this study are available through the Norwegian Research Data Archive/Bjerknes Climate Data Centre and can be accessed under https://doi.org/10.11582/2022.00012.

**Author contribution**

J.S. conceived the study, designed the model experiments, performed the model simulations, analyzed and interpreted the model data, and wrote the manuscript with contributions from all co-authors. A.A. post-processed model output data. A.A., N.J.S., and H.L. contributed to the model data analysis and interpretation and to the writing of the manuscript.

**Competing interests**

The authors declare no competing interests.

**Acknowledgements**

All authors acknowledge funding from the Research Council of Norway (project IMPOSE, grant 294930) and from the Bjerknes Centre for Climate Research (project LOES). J.S. has received funding from the European Union's Horizon 2020 680 research and innovation program under grant agreement No 820989 (project COMFORT) and grant agreement No 869357 (project OceanNTs). The work reflects only the authors' view; the European Commission and their executive agency are not responsible for any use that may be made of the information the work contains. Supercomputing and storage resources were provided by UNINETT Sigma2 (projects nn9708k/ns9708k).

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

**Table 1: Overview of simulations without ($B$) and with ($OS$) overshoot.**

| Simulation | Parent simulation | Cumulative positive emissions | Cumulative CDR | Time between positive and negative emissions | Ensemble members |
|---|---|---|---|---|---|
| $B^{1500}$ | - | 1500 Pg C | - | - | 3 |
| $B^{1750}$ | - | 1750 Pg C | - | - | 3 |
| $B^{2000}$ | - | 2000 Pg C | - | - | 1 |
| $B^{2500}$ | - | 2500 Pg C | - | - | 3 |
| $OS_0^{250}$ | $B^{1750}$ | 1750 Pg C | 250 Pg C | 0 years | 3 |
| $OS_{100}^{250}$ | $B^{1750}$ | 1750 Pg C | 250 Pg C | 100 years | 3 |
| $OS_0^{500}$ | $B^{2000}$ | 2000 Pg C | 500 Pg C | 0 years | 1 |
| $OS_{100}^{500}$ | $B^{2000}$ | 2000 Pg C | 500 Pg C | 100 years | 1 |
| $OS_0^{1000}$ | $B^{2500}$ | 2500 Pg C | 1000 Pg C | 0 years | 3 |
| $OS_{100}^{1000}$ | $B^{2500}$ | 2500 Pg C | 1000 Pg C | 100 years | 3 |

**Table 2: SAT evolution and AMOC decline in the *B*-simulations with 100 years of positive emissions and 300 years of zero emissions thereafter. Values are ensemble means (except for $B^{2000}$) and the ranges given are minimum and maximum values for individual ensemble members.**

| Simulation | Peak SAT* | SAT decline after peak* | Final SAT# | AMOC decline*+ |
|---|---|---|---|---|
| $B^{1500}$ | 1.86°C (1.85 - 1.95) | 0.59°C (0.60 - 0.75) | 1.70°C (1.60 - 1.78) | 11.0 Sv (48.4%) (10.6 - 12.0; 46.7 - 52.7%) |
| $B^{1750}$ | 2.20°C (2.15 - 2.26) | 0.66°C (0.57 - 0.81) | 2.09°C (2.07 - 2.13) | 13.5 Sv (59.5%) (13.1 - 14.2; 57.6 - 62.5%) |
| $B^{2000}$ | 2.35°C | 0.63°C | 2.37°C | 15.8 Sv (70%) |
| $B^{2500}$ | 3.01°C (2.99 - 3.08) | 0.39°C (0.40 - 0.55) | 3.38°C (3.31 - 3.43) | 17.3 Sv (76%) (17.2 - 17.6; 75.7 - 77.5%) |

*an 11-year running mean has been applied before calculating mean and ranges

#average over the last 11 years of the simulations (years 390-400)

+pre-industrial AMOC strength at 40°N is 22.7 Sv in NorESM2-LM

**Table 3: Reversibility of key aspects of the Earth system, based on an 11-year average at the end of the reference and overshoot simulations (years 390-400). Note that this definition implies that we compare REV$_{95}$ for the short overshoots and REV$_{195}$ for the long overshoots. Shown is the mean over three ensemble members (if available), and, for the reference simulation $B^{1500}$ in parentheses, the range of internal variability as defined in the text. Bold font indicates reversibility, normal font indicates irreversibility for a given variable and overshoot.**

| Simulation | ΔSAT | ΔCO$_2$ atm. | Steric sea level rise | AMOC | ΔPP | ΔpH | ΔO$_2$ | ΔNBP | Δ permafrost area | ΔC permafrost |
|---|---|---|---|---|---|---|---|---|---|---|
| $B^{1500}$ (reference) | 1.70°C (1.54 to 1.86) | 187.8 ppm (186.9 to 188.8) | 39 cm (38 to 40) | 20.5 Sv (19.8 to 21.1) | -1.45 Pg C yr$^{-1}$ (-1.91 to -0.99) | -0.1035 (-0.1037 to -0.1032) | -1.29 × 10$^4$ Tmol (-1.33 to -1.25) | -0.13 Pg C yr$^{-1}$ (-0.74 to 0.48) | -5.76 × 10$^6$ km$^2$ (-6.08 to -5.43) | -26.9 Pg C (-28.5 to -25.3) |
| $OS_0^{250}$ | **1.80°C** | **187.6 ppm** | 42 cm | **20.3 Sv** | **-1.55 Pg C yr$^{-1}$** | -0.1046 | -1.38 × 10$^4$ Tmol | **-0.24 Pg C yr$^{-1}$** | **-5.88 × 10$^6$ km$^2$** | -29.1 Pg C |
| $OS_{100}^{250}$ | **1.77°C** | 185.1 ppm | 43 cm | **19.8 Sv** | **-1.55 Pg C yr$^{-1}$** | -0.1064 | -1.44 × 10$^4$ Tmol | **-0.19 Pg C yr$^{-1}$** | **-5.95 × 10$^6$ km$^2$** | -31.5 Pg C |
| $OS_0^{500}$ | **1.64°C** | 184.7 ppm | 41 cm | **19.9 Sv** | **-1.19 Pg C yr$^{-1}$** | -0.1046 | -1.35 × 10$^4$ Tmol | **-0.16 Pg C yr$^{-1}$** | **-5.56 × 10$^6$ km$^2$** | **-25.9 Pg C** |
| $OS_{100}^{500}$ | **1.63°C** | 180.9 ppm | 45 cm | **20.0 Sv** | **-1.43 Pg C yr$^{-1}$** | -0.1084 | -1.48 × 10$^4$ Tmol | **-0.07 Pg C yr$^{-1}$** | **-5.48 × 10$^6$ km$^2$** | -31.9 Pg C |
| $OS_0^{1000}$ | **1.75°C** | **187.7 ppm** | 46 cm | 19.7 Sv | **-1.55 Pg C yr$^{-1}$** | -0.1078 | -1.53 × 10$^4$ Tmol | **-0.08 Pg C yr$^{-1}$** | **-5.77 × 10$^6$ km$^2$** | -49.1 Pg C |
| $OS_{100}^{1000}$ | 1.96°C | 184.8 ppm | 54 cm | **20.9 Sv** | **-1.74 Pg C yr$^{-1}$** | -0.1147 | -1.80 × 10$^4$ Tmol | **-0.03 Pg C yr$^{-1}$** | -6.17 × 10$^6$ km$^2$ | -67.3 Pg C |

880