# Peer review of "Emit now, mitigate later? Earth system reversibility under overshoots of different magnitude and duration"

_Earth System Dynamics, 2022_

## Author Comment (AC1)

**Authors' response to the review by Fortunat Joos**

We would like to thank Fortunat Joos for the positive evaluation of our manuscript and for the constructive and helpful criticism. We have revised our manuscript and addressed all points that have been raised as detailed in our point-by-point response below (original comments in grey, italic font). Proposed verbatim changes or additions to the manuscript are highlighted in this colour.

*The authors present results from a comprehensive set of 400-year long, idealized emission-driven climate-carbon simulations with the Norwegian Earth System Model to investigate the impacts of carbon dioxide removal (CDR) and a temporary exceedance of 2°C warming for the global carbon budget, Earth system parameters such as global surface air-temperature, steric sea level rise, ocean heat content, Atlantic Meridional Overturning, ideal watermass age, ocean oxygen, pH, and productivity, permafrost extent and carbon stocks, and climate metrics such as TCRE.*

*In the reference simulations, CO2 emission increase and decrease and are phased out over the course of the simulation to a total of 1500 GtC. In overshoot scenarios, prescribed CO2 emissions are temporarily larger than in the reference simulation and these "excess" emissions are later compensated by negative emissions (net carbon dioxide removal (CDR)) to yield again cumulative emissions of 1500 GtC. The authors apply excess emissions/CDR of 250, 500, and 1000 GtC and two different timing for CDR in their setups.*

*The authors compare results between the reference simulation and the over-shot scenarios, all with equal total cumulative emissions, to investigate whether climate parameters and the carbon budget (atm, ocean, land biosphere C stocks) of the overshoot simulations approach those simulated in the reference towards the end of the 400-yr simulations. The authors denote a parameter change (e.g., Delta-SAT) reversible if the parameter value in the overshoot scenario is equal to that of the reference simulation within the bounds of internal, interannual variability for the last 11 years of the simulation. In other words, reversibility is only investigated for the amount of excess emissions/CDR applied – here 250 GtC in the "realistic case", but not for the total anthropogenic CO2 emissions of 1500 to 2500 GtC.*

*Overshoot scenarios have been investigated in earlier studies, however, with Earth System Models of Intermediate Complexity. Here, the authors apply a state-of-the-art, comprehensive Earth System which also includes a module to represent permafrost extentand permafrost carbon. The authors apply the model in 11 different setups (control plus 10 scenarios) with some ensemble members and each simulation for a 400-year period. This represents a considerable effort in terms of computing time and human resources. The results are very interesting and comprehensively assessed for a broad range of global parameters. The manuscript is generally clear and well written. I recommend publications after the following comments have been taken into account.*

*The comments should not require any new simulations or analyses, but, in my opinion, the text must be carefully revised in the abstract, introduction, and conclusion sections and adjusted elsewhere, to avoid misleading sentences and possible misinterpretations.*

Thank you for the positive evaluation.

1) *My main concern is related to how reversibility and irreversibility are defined and conclusions are framed. The concept of (ir)reversibility is applied in a very restricted sense in this study. As noted above, reversibility is only investigated for the extra emissions (above the reference case) and for later removal of these extra emissions by CDR, but not for total anthropogenic emissions. (This is o.k. per se, but must be more clearly explained in the abstract and elsewhere).*

*Earlier studies have addressed reversibility relative to total anthropogenic carbon emission and found substantial irreversibility for atmospheric CO2, surface air temperature, ocean acidification and deoxygenation and other variables on multicentennial time scales. For example, ESM simulations where emissions are suddenly stopped in the year 2000 or 2100 reveal large climate inertia and irreversibility (e.g., (Frölicher and Joos, 2010; Joos et al., 2011). Similarly, a CO2 pulse response model intercomparison study documents the long life time of atmospheric CO2 and climate perturbations and thus irreversibility in a hierarchy of models (Joos et al., 2013).*

*Under the authors' narrow definition, reversibility is found for overshoots/CDR of up to 500 GtC and variables concerning the fast-reacting climate components such as surface air temperature and ocean surface pH. Properties related to the deeper ocean (steric sea level rise, OHC, deep ocean temp., oxygen, pH) and permafrost show irreversibility even under the authors' definition. Irreversibility for most variables is found for large overshoot/CDR (1000 GtC). The authors judge CDR of up to 250 GtC as "realistic" (e.g., around L 550). Thus, any emissions exceeding 250 GtC have an impact on SAT, CO2, .. that is irreversible on human, centennial time scales.*

*The sentence in the abstract: "Many aspects of the Earth system including global average surface temperature, marine and terrestrial productivity, strength of the Atlantic meridional overturning circulation, surface ocean pH, surface O2 concentration, and permafrost extent are reversible on a centennial time scale except in the most extreme overshoot scenario considered in this study" is very misleading. If cited out of context, the results of this study can be misused.*

*In brief, the authors must reword the abstract and conclusion sections and clearly explainin the introduction that reversibility/irreversibility in this study is restricted to the (relatively small) amount of overshoot emissions/CDR and that CO2 emissions (exceeding what can be plausibly removed by CDR) cause irreversible climate change.*

Thank you for making us aware of this problem. We agree that a more careful wording regarding the term (ir)reversibility is needed to avoid misunderstandings or even misuse of our results. In a revised version of our manuscript, we made changes to the abstract, the introduction, and conclusions as follows:

We rewrote the beginning of the **abstract** as follows: "Anthropogenic $CO_2$ emissions cause irreversible climate change on centennial to millennial time scales, yet current mitigation efforts are insufficient to limit global warming to a level that is considered safe. Carbon Dioxide Removal (CDR) has been suggested as an option to partially reverse climate change and to return the Earth system to a less dangerous state after a period of temperature overshoot. Whether or to what extent such partial reversal of climate change under CDR would happen is, next to socio-economic feasibility and sustainability, key to assessing CDR as a mitigation option. Here, we use…"

We reworded the sentence "Many aspects of the Earth system […] are reversible on a centennial time scale…" to read: "The legacy of an overshoot is, on a centennial time scale, indiscernible (within natural variability) from a reference case without overshoot for many aspects of the Earth system including global average surface temperature, marine and terrestrial productivity, strength of the Atlantic meridional overturning circulation, surface ocean pH, surface $O_2$ concentration, and permafrost extent except in the most extreme overshoot scenario considered in this study."

We further eliminated the word "irreversibility" from the last sentence of the abstract by rewording: "Hence, the effectiveness of CDR in partially reversing large scale patterns of climate change might not be the main issue of CDR…"

In the **Introduction**, we begin the second paragraph as follows: "From an Earth system perspective, the issue of reversibility (used here to denote a partial reversal of climate change in an overshoot

pathway towards an Earth system state in a reference pathway without overshoot) is key in assessing the effectiveness and risks of mitigation pathways that rely on carbon dioxide removal (CDR):…"

We further modified the first sentence of the 6[th] paragraph to read: "While it is well established that $CO_2$ emissions that remain in the Earth system cause irreversible climate change (e.g., Solomon et al. 2009; Frölicher and Joos, 2010; Joos et al., 2011), the term "(ir)reversibility" of climate change has also been used to describe whether (or not) the climate system will, if $CO_2$ is removed from the atmosphere, return to a reference state where no CDR has been applied."

In the **Conclusions** section, we modify the second sentence as follows: "To assess whether climate change can be partially reversed by CDR, we compare…", thereby removing the term "reversibility". After this sentence, we add a clarification of how the term "reversibility" in our study is to be understood: "We define an aspect of the Earth system to be reversible through application of CDR if the mean state after an overshoot is within the internal variability of the reference case without overshoot. We stress that this definition neither implies reversibility in the absence of CDR nor reversibility of climate change that is committed to in the reference scenario."

To avoid the wording that "many aspects of the Earth system are reversible", we rewrote the first sentence of the third paragraph of the conclusions section as follows: "We further find that, on a timescale of 100 to 200 years after all emissions cease, CDR is effective in partially reversing the global mean state of many aspects of the Earth system…"

> 2) *The NoESM has a low TCR and a low TCRE compared to observation-constrained estimates (e.g., Sherwood et al., 2020; Steinacher and Joos, 2016). Irreversibility and hysteresis has found to be higher for higher TCR and TCRE. The authors should discuss this aspect and consider it when making statement about reversibility.*

We have added this aspect to section 3.2, when first discussing reversibility in the overshoots: "We note that previous studies with intermediate complexity ESMs (MacDougall 2015; Jeltsch-Thömmes et al. 2020) have shown that hysteresis and irreversibility generally increase with increasing climate sensitivity. Therefore, owing its low climate sensitivity, NorESM2 most likely shows a relatively high degree of reversibility compared to higher sensitivity ESMs."

We also have added a discussion on this in the first paragraph of the conclusions section as follows: "We also note that our Earth system model (NorESM2) has a low climate sensitivity, and that previous studies with intermediate complexity ESMs (MacDougall 2015; Jeltsch-Thömmes et al. 2020) have shown that hysteresis and irreversibility are generally smaller for low climate sensitivity."

*Further comments*

*Abstract and L558: ".. it seems that reversibility might not be the main concern" This text must be reworded. First, this statement hinges on the very narrow definition of reversibility applied in this study. (ir)reversibility in this study is only assessed for the carbon emissions that are later removed by CDR. It is only the (ir)reversibility for these "overshoot" emissions that are discussed in this study. The authors estimate that up to 250 GtC may be removed by CDR and that larger removal is likely unrealistic. Thus, irreversibility for most of the emissions - 1500 to 2500 GtC in their baseline scenarios – is not assessed at all. This must be clearly stated in the abstract, conclusion, and elsewhere. Further, this statement is also subject to the low TCR and TCRE of the NorESM.*

We reworded this sentence in the abstract as described above. We believe that with the major changes to the text in the abstract, introduction and conclusions (and also elsewhere) as summarized above the usage of the terms (ir)reversibility and limitations of our study (low TCR model) should be clarified.

*Methods: I suggest giving some more information in the method section on how permafrost is modelled and whether changes in vegetation distribution and associated feed backs are considered.*

We have added this information to the methods section: "NorESM2 employs version 5 of the Community Land Model (CLM5, Lawrence et al. 2019), which is capable of simulating key thermal, hydrologic, and biogeochemical processes associated with permafrost and their response to climate change. Compared to previous model versions, CLM5 includes several improvements (increased soil depth, improved vertical resolution particularly in the top 3 m, vertically resolved soil biogeochemistry, and changes to modelled snow density, among others) enabling more realistic modelling of permafrost and active layer dynamics (Lawrence et al. 2019). The permafrost region is defined here as the geographic area where the model simulates a maximum active layer thickness shallower than 3 m. We note that CLM5 does not simulate the spatial dynamics of vegetation cover and competition between different plant functional types. For example, northward expansion of plant species due to climate warming is not represented. Changes in carbon stocks associated to vegetation are therefore owed to changes in the plant carbon metabolism response to variations in atmospheric $CO_2$ and temperature only."

*L113: Please give also an estimate of the model's TCRE and how this value compares to the TCRE of other models or as constrained in probabilistic approaches.*

We have added this information as follows: "Consistent with the relatively low TCR, NorESM2 also shows a low transient climate response to cumulative emissions (TCRE) of 1.32°C EgC$^{-1}$ (CMIP6 range 1.32-2.30; Arora et al. 2020) and therefore shows relatively little warming for a given amount of $CO_2$ emissions."

*Line 121: It would be clearer to delete "in total" or say "Each of the six .." or similar*

This has been done.

*157: Please do not claim that the scenarios are designed to meet the Paris temperature goal(s). The Paris Agreement says: "keep global warming well below 2°C and to pursue efforts to limit it to 1.5°C". The wording "keep below" excludes, at least in my interpretation, warming equal or larger than 2°C. All scenarios, except the reference B-1500, exceed the 2°C limit and none of the scenarios meets the 1.5°C limit as shown in Fig. 1. Please reformulate the text (e.g. "Our scenarios are designed to eventually meet the 2°C limit mentioned in the Paris Agreement after a period of overshoot (Delta-T> 2°C) ..").*

Thank you for making us aware of this. We reformulated our sentence as suggested: "Our experiment design aims at simulating, in an idealized fashion, emission pathways that eventually meet the temperature target mentioned in the Paris Agreement of keeping global warming to well below 2°C after a period of overshoot (with global warming >2°C),…"

*L171ff: The explanations on internal variability and the definition of reversibility are not fully clear. (I apologize that the comment is a bit picky. However, it would be good to be very clear on these concepts and how they are used)*
  - *(L173-174) Please define more precisely how variances are computed. Do you use annual-average or monthly-average data or model output for each time step or something else to compute variances? Are variances computed from spatially-averaged time series to estimated*

*internal variability for spatially averaged parameters or are variances first computed at each grid cell and then averaged? I guess you use annual data and compute variance from spatially-averaged time series.*

Yes, this is correct, we use *annual data and compute variance from spatially-averaged time series.* In the revised version of our manuscript this will be clarified (see revised text below).

- *(L174-175) In the time of emergence approach cited by the authors, it is typically assumed that the threshold is 2\*N/S, where S is the signal and N the noise. N is typically assumed to equal the standard deviation (sdv). Here the signal is compared with "the square root of the summed variances" In other words, the average variance from the three ensembles is multiplied with sqrt(3)=1.7. The authors may note that this factor 1.7 is slightly less than the 2\*sdv used by others, but more conservative in the current application.*

We see that our description of how we define internal variability was not sufficient in providing the details. In fact, we use a relatively conservative (for our application) threshold of $1\sigma$ to define internal variability. We have clarified this in our revised manuscript by adding an equation for our definition of internal variability (see below). We also mention that the $1\sigma$ threshold used is more conservative for our application (please see revised text below).

- *(L171) "the ensemble mean of the overshoot simulation returns to the reference pathway within the range of internal variability". It may be good to say already here that 11-year averages from the overshoot and reference run are compared.*

This has been done, see revised text below.

- *L171: The climate system may also feature internal variability on time scales longer than interannual or seasonal variability. Do you assume in your approach that such longer-scale variability is always smaller than interannual variability?*

We do not make this assumption here (for some variables like AMOC, decadal variability is certainly large). By averaging over three ensemble members (both for the reference simulation and the overshoot) we remove a considerable part of decadal and longer-term variations. We mention this in our revised manuscript, see revised text below.

To address the various points regarding our definition of internal variability, we have revised our manuscript as follows: "For a given point in time *t* we calculate the variance of an annually and spatially averaged time-series $x_i(t)$ for ensemble member *i* (*i*=1...3) over 11-years, centred at point *t*, and define internal variability (IV) as the square root of the mean of the variances of the 3 ensemble members:

$$\text{IV}(t) = \sqrt{\frac{1}{3}\sum_{i=1}^{3}\frac{1}{11}\sum_{k=-5}^{5}(x_i(t_k) - \bar{x}_i)^2}$$

This definition is similar to defining internal variability as the standard deviation ($1\sigma$) of a time-series over a period of 33 years, except that we compensate for a shorter time-interval by using several ensemble members. The 11-year sliding window used here is consistent with the fact that we present most of our results as moving averages with the same window length (11 years). Defining internal variability as one standard deviation ($1\sigma$) of interannual variations leads to relatively conservative threshold for reversibility, compared to some previous studies, which used a $2\sigma$ definition of internal variability (e.g., Keller et al., 2014). Decadal and longer-term variability, which can be larger than interannual variations for some variables, is partly removed in our approach by averaging over three ensemble members."

*L314: annual mean or annual maximum value of active layer depth?*

We clarified this by adding "annual maximum".

*L319: "which show that the physical state of permafrost is reversible under temperature reduction" Please add "in our model". Do you think that formation of thermokarst lakes, coastal zone erosion, and other landscape changes are adequately treated in your model to make a strong conclusion on permafrost reversibility? Do you consider dynamic vegetation changes in these simulations or is vegetation prescribed? Are albedo feedbacks associated with the potential transition from forest to tundra included in the model? Please elaborate on these points here and/or in the discussion.*

That physical permafrost area is recovering under CDR is not only seen in our model. We agree, however, that this sentence should be formulated more carefully. We have therefore revised this sentence as follows: "…which show that the physical extent of permafrost area mainly follows the SAT trajectory and tends to recover under temperature reduction (Boucher et al. 2012; MacDougall 2013; Lee et al. 2019)." We have also added a sentence mentioning the limitations of our model regarding small scale features of permafrost degradation: "It is, however, worth mentioning that landscape changes and hydrological responses to permafrost thaw, such as coastal erosion, excess ice melting, and formation of thermokarst lakes are highly heterogeneous and depend on small scale processes that are neither resolved nor parameterized in our model. Therefore, irreversible changes at the (unresolved) landscape scale would occur even if the modelled large-scale physical state of the soil is found to be reversible according to our definition." Please see next point for our response on vegetation/albedo feedbacks.

*L321: What about vegetation distribution, e.g. extent of different ecosystems such as boreal forest, tundra, tropical forests?*

We have added information on this: "Note that the distribution of vegetation is prescribed in our model, such that (potentially irreversible) changes in vegetation carbon as well as biophysical feedbacks (changes in albedo and roughness length) caused by shifts in vegetation composition are most likely underestimated in our simulations. For example, a northward tree-line expansion or shrubification in high latitudes due to climate warming cannot be captured by our model."

*L330: Fig 9h -> Fig. 6h?*

Thank you, this has been corrected.

*L369: typo: "As a mentioned"*

Thank you, this has been corrected.

*L394: I find the following statement and the text on line 394 to 399 misleading: "We note, however, that negative emissions are indeed effective in reducing the rate of sea level rise to a value similar or lower than that of the reference simulation (Fig. 8c)." 8c only shows rates of sea level rise for the last 50 years. Earlier rates of change are, however, larger for the overshoot cases than for the reference case without overshoot. This should be made clear. Please add an additional panel in Fig. 8 showing the maximum rates of sea level rise over the course of each 400-yr long simulation.*

We agree that the sentence as it stands is misleading. We have reworded this sentence as follows: "We note, however, that negative emissions are indeed effective in reducing the rate of sea level rise after an overshoot to a value similar or lower than that of the reference simulation (Fig. 8c)." We will

also add a panel in Fig. 8 showing the rate of sea level change over the course of the 400-year simulation.

*L490 ff on tipping points: One could mention that the findings here are broadly in agreement with results from emission-driven EMIC studies or emission-driven ESM studies. It may also be interesting to mention the study by (Kleinen et al., 2020) who found methane to respond strongly to warming..*

Thank you for this suggestion. In our revised manuscript we have added the sentence "This result is broadly consistent with previous model studies using intermediate complexity ESMs (e.g., Steinacher and Joos 2016, Jeltsch-Thömmes et al. 2020) or CMIP6 ESMs (e.g., Koven et al. 2022)."

Regarding the methane emissions, we have added the sentence "Likewise, methane emissions from wetlands, which have been found to react strongly to warming (Kleinen et al. 2020), do not affect atmospheric methane concentrations in our model configuration." to the conclusions section.

*L558: What is a "realistic overshoot scenario"? Do you mean a scenario with only a limited amount of CDR (< 250 GtC). Please say so explicitly.*

This has been done.

---

## Author Comment (AC2)

**Authors' response to the review by Kirsten Zickfeld**

We would like to thank Kirsten Zickfeld for the positive evaluation of our manuscript and for the constructive and helpful criticism. We have revised our manuscript and addressed all points that have been raised as detailed in our point-by-point response below (original comments in grey, italic font). Proposed verbatim changes or additions to the manuscript are highlighted in this colour.

*Schwinger et al investigate the reversibility of the Earth system for a range of idealized scenarios that differ with regard to the amount and duration of overshoot. They consider a change to be reversible if the modelled ensemble mean of an overshoot simulation returns to the reference simulation within the range of internal variability. Consistently with earlier studies they find that most Earth system changes are reversible, except for aspects with longer response timescales, such as permafrost carbon and seawater properties of the deep ocean. They do not identify tipping points in their simulations following overshoot.*

*The manuscript by Schwinger et al. is a valuable contribution to a relatively small body of literature that investigates Earths system reversibility in emissions-driven simulations, allowing for feedback between climate and the carbon cycle. The manuscript is well written, the methodology is adequate and sufficiently documented, and the conclusions are supported by the findings. There are a few instances where the manuscript would benefit from additional explanations, or definition of terminology for an interdisciplinary readership.*

Thank you for this positive evaluation.

*Specific comments:*

*l.185: Comparing reversibility for a given year has the disadvantage that it does not allow for a clean separation of the effect of overshoot duration, as differences could also be due to shorter time left to adjust to the final forcing level in the longer vs. shorter overshoot simulations.*

Yes, this is true. This point is most relevant for the oceanic zonal mean sections shown in Fig. 10. We have added a figure to the Appendix (Fig. A4) showing the reversibility 95 years after the negative emissions cease (REV$_{95}$) for the short overshoot. This allows a clean comparison for the effect of overshoot duration when compared to Fig. 10. We have added a few sentences discussing this new figure at the end of section 3.7 as follows: "So far, we have chosen to assess reversibility for the same simulation year, that is, we compare REV$_{95}$ for the long overshoot simulations (95 years after negative emissions ceased, Fig. 10) to REV$_{195}$ for the short overshoot simulations (195 years after negative emissions cease, Figs. A2 and A3). To allow for a clean comparison of the effect of overshoot length on reversibility, Fig. A4 shows REV$_{95}$ for the short overshoots (i.e., reversibility derived from simulation years 290-300). Comparing Fig. A4 and Fig. 10 reveals that the volume of water masses showing irreversible change tends to be larger in the long overshoots, although the spatial patterns are broadly similar. The main difference between the short and long overshoots is that the irreversible changes in the deeper ocean are much more pronounced for the long overshoot duration, indicating a clear benefit of limiting the duration of an overshoot."

*l. 265: It could be pointed out that atmospheric CO2 in the overshoot simulations temporarily "undershoots" CO2 levels in the reference simulation.*

We followed this suggestion by adding the sentence *"We note that during the negative emission phases, atmospheric $CO_2$ concentrations decrease below the concentration in the reference simulation in all overshoots. This effect is largest (about 50 ppm) for the large overshoots, which have the fastest rate of CDR."*

*l. 283: Before discussing fractional quantities in Fig. 4 I suggest to discuss the cumulative fluxes (Fig. 3), which don't have the denominator changing at the same time and are therefore more intuitive. Here the reason for the decline in the cumulative fluxes during the negative emissions phase could be explained (e.g. the reversal in pCO2 gradient mentioned in l. 387-388.).*

We followed this suggestion and reworded/extended the beginning of this paragraph: "During phases with negative emissions in the overshoot simulations, both land and ocean become a source of $CO_2$ such that their carbon stocks are reduced (Fig. 3c,d). For the ocean this happens as soon as the $CO_2$ partial pressure difference between atmosphere and ocean becomes negative. For the land, a reduced $CO_2$ fertilization effect shifts the overall balance between carbon uptake through net primary production and carbon release through heterotrophic respiration towards the latter. However, since these processes are slow and lag the reduction of the cumulative total of emissions, there is a rapid increase of $CF_O$ and $CF_L$ (Fig. 4b,c)."

*l. 350: Section 3.5: I suggest to define and explain the meaning of the biogeochemical quantities discussed in this section (preformed vs. remineralized carbon, AOU etc.) to make sure the findings are accessible to an interdisciplinary readership.*

We have added text explaining these biogeochemical quantities as follows: "The main carbon reservoirs considered here are vegetation carbon, permafrost carbon, and non-permafrost soil carbon for land, as well as remineralised and preformed dissolved inorganic carbon (DIC) for the ocean. Changes in permafrost carbon are obtained as cumulative carbon fluxes summed over all permafrost grid cells, i.e. those grid cells that are defined as permafrost in the pre-industrial control simulation. Preformed DIC originates from atmospheric $CO_2$ dissolves in the surface ocean and is transported into the interior by ocean circulation. In contrast, remineralized DIC has been transported into the interior ocean through the biological carbon pump: Biological uptake by planktonic organisms near the ocean surface, sinking to depth as particulate organic matter, and subsequent remineralization by bacterial activity. We note that the remineralization of organic carbon consumes oxygen (if present in sufficient quantity), such that oxic remineralization can be measured by apparent oxygen utilization (AOU), defined as the oxygen deficit in a water parcel relative to its saturated oxygen content."

*l. 377: Mention that inclusion of vegetation dynamics could affect reversibility.*

This has been added as follows: "We note that the inclusion of vegetation dynamics in our model would most likely affect these results, since changes in biogeography would lead to larger changes in land carbon pools and larger time lags between drivers and response."

*l. 387-388: This is the first time this is mentioned. I suggest to discuss this earlier (e.g. in section 3.3.).*

This has been done. See our responses to the 2nd and 3rd specific comment above.

*l. 390: Irreversibility of thermosteric sea level rise was also investigated in Ehlert & Zickfeld, 2018, https://doi.org/10.5194/esd-9-197-2018.*

We have included this paper in the list of cited works.

*l. 395 – 398: I don't follow this argument. Perhaps the amount of sea-level rise corresponding to a 1.5C global warming limit was "implicitly accepted", but not the additional sea level rise resulting from overshoot of the warming limit?*

We agree that this was not very well explained. Our argument is that the additional sea level rise due to an overshoot remains relatively small compared to the amount of sea level rise committed to in the reference simulation (< 20% except for the most extreme overshoot). We make this clear by rewording: "The additional steric sea level rise due to an overshoot remains relatively small compared to the sea level rise committed to in the reference simulation in our model (<20% at year 400 except for the most extreme overshoot $OS_{100}^{1000}$). Thus, the rate of sea level rise determines the pace and cost of necessary adaptation for the decades to centuries after an overshoot. Therefore, limiting the rate of sea level rise after an overshoot might arguably be more policy relevant in the context of negative emissions than a relatively limited contribution of the overshoot to the sea level rise itself. "

*Figure presentation: Vertical lines showing the positive emissions, negative emissions and zero emissions phases could be included. Also, it would be helpful to have the legend repeated in figures where changes are non-monotonic as a function of overshoot size or duration (e.g. Figs. 5 and 6).*

We have added shadings in the background of each time-series figure indicating the different phases of our simulations. We have also added legends to figures 5 and 6 as suggested.